# Host oxidative stress primes mycobacteria for rapid antibiotic resistance evolution

Evan Pepper-Tunick [1,2], Vivek Srinivas [1], Fred D. Mast [3,4], Song Li[3], Sagan Russ [1], Weston Hanson [1], Amy D. Zamora [1], Wei-Ju Wu[1], Matthew Silcocks [5], Dang Thi Minh Ha[6], Sarah J. Dunstan [5], Thuong Nguyen Thuy Thuong [7,8], Serdar Turkarslan [1], John D. Aitchison [3,4,9], Mario L. Arrieta-Ortiz [1] ✉ & Nitin S. Baliga[1,2,10,11,12,13] ✉

The rapid emergence of multidrug-resistant *Mycobacterium tuberculosis* (Mtb) threatens global tuberculosis (TB) control, yet the mechanisms enabling rapid evolution of resistance in Mtb remain poorly understood. Here, we show that pre-existing mutations in oxidative stress response genes create permissive genomic backgrounds that accelerate high-level isoniazid resistance (INH^R), challenging the paradigm that resistance mutations must precede compensatory adaptation. Using *Mycobacterium smegmatis* mc²155 (Msm) as a model, we demonstrate that brief exposure to sublethal isoniazid (INH) enriches for "low-level resistance and tolerance" (LLRT) mutants in a single step. LLRT mutants, particularly those with *ohrR* loss-of-function mutations, acquire high-level resistance (>500× IC₅₀) at ~6-fold higher rates than wildtype, primarily through otherwise deleterious mycothiol biosynthesis mutations that become tolerable in an oxidative stress-buffered background. Crucially, sublethal oxidative stress alone, mimicking host immune pressure, nearly tripled the rate of INH^R evolution. Analysis of 1578 clinical Mtb isolates revealed significant enrichment of oxidative stress-related loci among those associated with INH^R. Reanalysis of genome-wide CRISPRi data further linked oxidative stress response pathways to survival under multiple antibiotics. Together, these findings suggest that host-imposed oxidative stress and suboptimal drug exposure may prime Mtb populations for rapid resistance evolution, highlighting oxidative stress defenses as potential targets to limit resistance emergence.

The rise in drug-resistant *Mycobacterium tuberculosis* (Mtb) infections, the bacterium responsible for tuberculosis (TB), is a growing global health crisis[1], creating an urgency to understand the mechanisms driving rapid gain of antimicrobial resistance in mycobacteria. Because antibiotics target essential functions, such as cell wall synthesis, DNA replication, transcription, translation, or metabolism, resistance conferring mutations often carry a fitness cost typically resulting in a decreased growth rate of the microbe in the absence of the selective

pressure[2–4]. Despite the associated fitness cost, the introduction of new antimicrobial compounds is often immediately followed by the emergence of clinically resistant strains[5,6]. Compensatory adaptations can ameliorate the fitness cost of the resistance conferring mutations through a number of mechanisms, including epistatic interactions, gene duplications, the use of alternative pathways to reduce the need of the drug target, and many others[7–9]. In the presence of a strong selective pressure, resistant cells will outcompete their susceptible

---

cousins and eventually dominate the population, especially if selective pressures remain constant[10,11]. However, in natural environments, including both clinical and agricultural settings, the strength and type of forces that determine which mutant alleles are selected for or against usually fluctuate[12]. Bacterial pathogens may be exposed to low concentrations of antibiotics for extended periods of time, allowing low-cost mutations that give some survival advantage an opportunity to be enriched and subsequently selected[13–17]. For example, in vivo and in silico pharmacokinetic studies have suggested that inadequate antibiotic penetration and accumulation into the granulomas of TB patients can be common, and may occur for a number of reasons, including granuloma heterogeneity or treatment non-compliance[18–21]. The survival advantage conferred by mutations that precede full resistance (pre-resistance mutations) may be subtle, generating low-level resistance, tolerance, or other phenotypes that ultimately buy these cells time to propagate and acquire full, clinical resistance conferring mutations[22]. Despite significant progress in identifying and characterizing bacterial antibiotic-resistance mechanisms, we still do not fully understand how sub-inhibitory antibiotic exposure or host-induced stress triggers phenotypic resistance or tolerance. These conditions may also enrich pre-resistance mutations−variants that occur at relatively high frequencies because they carry low fitness costs.

Here, we report findings from an integrated strategy that leverages laboratory evolution using *Mycobacterium smegmatis* mc²155 (Msm) as a model mycobacterial species with parallel analysis of genotypic and phenotypic (antibiotic susceptibility) characteristics of clinical Mtb strains to identify pre-resistance mutations that precede and ultimately potentiate rapid gain of high-level resistance to the frontline TB drug isonicotinic acid hydrazide (isoniazid: INH). Our findings show that the mitigation of oxidative stress is a hallmark feature of pre-resistance in *Mycobacterium spp.*, and that mutations that enhance the ability to neutralize reactive oxygen species (ROS) create fertile grounds for selecting mutations conferring high-level INH resistance, without fitness tradeoff. Contrary to prior interpretations that compensatory adaptations appear later to ameliorate the cost of antibiotic resistance, our findings show that these adaptive changes are due to pre-resistance mutations that appear earlier, and accelerate the gain of high-level resistance-conferring mutations without the associated fitness tradeoff. Importantly, we discovered that brief pre-exposure of Msm to sub-inhibitory levels of ROS, to mimic host-induced stress, nearly tripled the rate of gain of high-level resistance to INH, and furthermore, that genes in the Mtb oxidative stress response (OSR) network were functionally associated in CRISPR interference (CRISPRi) screens with escape and survival from treatment with multiple frontline drugs. These findings help explain why Mtb rapidly acquires clinical resistance to new antibiotics and point to strategies that could slow the emergence of drug resistance.

## Results

### Brief one-step exposure to low-dose antibiotic enriches sub-populations of Msm with low-level resistance and tolerance to INH with no fitness trade-off

We investigated the consequence of brief exposure to a low-dose of antibiotic, as experienced by Mtb bacilli within a granuloma due to inadequate penetration of the drug, by subjecting eight replicate lines of log-phase Msm (mc²155) to 2× half-maximal inhibitory concentration (IC$_{50}$) (8.0 µg/mL) INH for 16 h. Following the brief treatment, culture aliquots were plated on 7H10 agar with 2× IC$_{50}$ INH and screened with ScanLag[23] to assess phenotypic heterogeneity of putative low resistance sub-populations of Msm (Supplementary Fig. 1). In parallel, aliquots from each replicate line were plated on antibiotic-free agar prior to the 16-hour liquid INH exposure and screened with ScanLag separately as untreated controls. Altogether, the growth of

632 INH-treated colonies and 537 untreated control colonies were detected and tracked with ScanLag. The 632 INH-treated colonies across the eight replicate lines clustered into three groups (Fig. 1a) based on principal component analysis (PCA) and k-means analysis of growth rate (Fig. 1b), time of appearance (Fig. 1c), and maximum colony size. Fifty-five colonies (at least six colonies from each replicate line), representative of phenotypic diversity across the three clusters, were characterized further to determine change in fitness in absence of antibiotic and level of resistance to INH (fold change in IC$_{50}$) relative to the wildtype ancestor. While there were no significant differences in average fitness and IC$_{50}$ of isolates from each cluster relative to the wildtype (Fig. 1d, e), at least 40 isolates exhibited low-level resistance (≥ 1.1× fold change IC$_{50}$ with respect to wildtype) with little-to-no fitness tradeoff (Fig. 1f). In summary, these findings demonstrate that low-level INH resistant sub-populations were enriched with brief exposure to low-dose antibiotic, and that these mutants co-exist within a larger naïve wildtype mycobacterial population, even in the absence of the antibiotic.

Genome re-sequencing and variant analysis suggested that the low-level resistance phenotypes of at least 15% (6/40) of all sequenced isolates may have emerged independently through point mutations in four genes: *ohrR*, *mfs1*, *ntaA_5* and *fas1* (Table 1, Source Data). Specifically, two isolates from the same line had a single base insertion resulting in frameshift at residue 4 (P4* *fs*) of *ohrR* (organic hydroperoxide reductase regulator). Loss of function mutations in *ohrR*, a transcriptional repressor of the *ohr* gene which encodes an organic hydroperoxide reductase, has been previously associated with low-level INH resistance in Msm[24,25]. One isolate with a nonsynonymous mutation in the major facilitator superfamily 1 gene, *mfs1* (G105D), exhibited ~1.8× fold increase in INH IC$_{50}$. MFS transporters, including the Mtb ortholog Rv2994, have been implicated in antibiotic efflux in many bacterial species[26,27]. Two isolates harbored a frameshift mutation at residue 51 (E50* *fs*) in *ntaA_5*, which encodes a putative xenobiotic compound monooxygenase from a class of enzymes implicated in detoxification of antibiotics, including tetracycline, rifampicin and imipenem[28–36]. Lastly, a sixth isolate with a nonsynonymous mutation (E177G) in *fas1*, which encodes fatty acid synthase 1 (FAS1), an enzyme required for de novo biosynthesis and elongation of fatty acids that ultimately become mycolic acids in the mycobacterial cell wall[37]. While FAS1 is yet to be associated with antibiotic resistance, the reaction products of FAS1 are reactants for the FAS II-system, mutations in which, including in InhA, are known to confer high-level resistance to INH in Msm and Mtb[38–42]. Additionally, genome re-sequencing of one untreated control colony per line from antibiotic free-plates identified only the *ntaA_5* variant and did not detect *ohrR*, *mfs1*, or *fas1* mutations. Although the sampling depth of untreated controls was limited, this result supports the interpretation that LLRT mutations pre-exist in the naïve populations and are enriched under sublethal INH exposure.

Prior work in *E. coli* has demonstrated that cyclic exposures to high-dose antibiotic, with an intermediate counterselection of fitness-compromising mutations, enriches sub-populations with phenotypic tolerance, which is eventually fixed through selection of tolerance-conferring mutations[22]. We performed time kill assays to investigate whether mutants selected with a single short-term exposure to low-dose INH also conferred tolerance to the antibiotic. We added 100× the wildtype minimum inhibitory concentration (MIC) of INH (1 mg/mL) to mid-log phase cultures of the parental wildtype mc²155 and each of the three mutants (excluding the *fas1*::E177G mutant), and assayed loss of survival over a 72-hour time course by counting colony forming units (CFUs) on 7H10 agar plates (Methods). The assay demonstrated that the three low-level resistant mutants were also significantly tolerant to INH, with their respective time kill curves never crossing the minimum duration for killing 99% (MDK$_{99}$) threshold, unlike the wildtype which reached MDK$_{99}$ within 20 h of treatment initiation (Fig. 1g). In

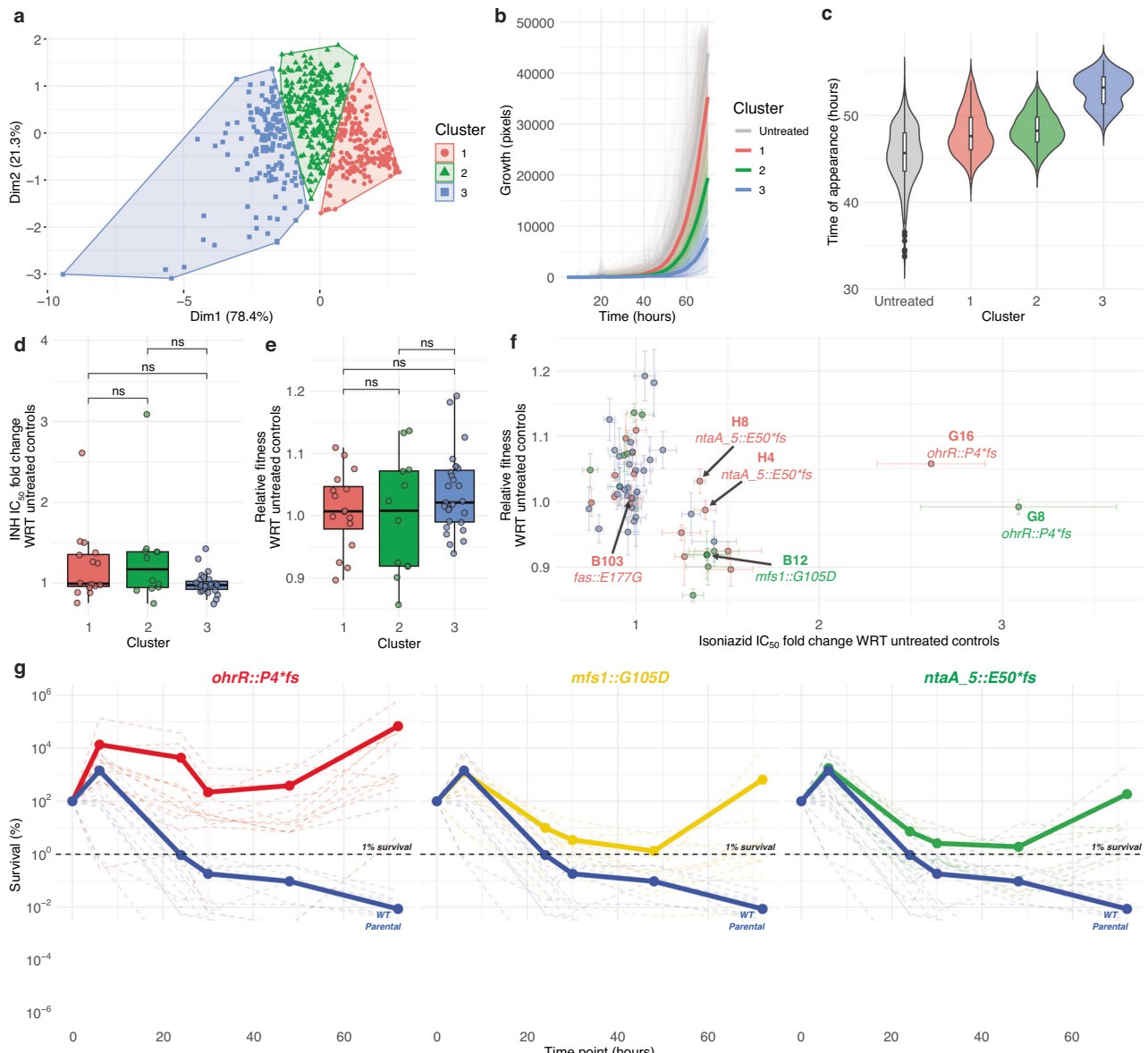

**Fig. 1 | Brief treatment with 2× IC$_{50}$ isoniazid enriches subpopulations with low-level resistance and tolerance, and no fitness tradeoff. a** Principal component analysis of the time of appearance, maximum growth rate, and maximum size of each colony on 7H10 agar with 2× IC$_{50}$ INH following 16 hours-exposure to 2× IC$_{50}$ INH in broth. Growth curves (**b**) and time-of-appearance (**c**) on 7H10 agar with 2× IC$_{50}$ INH for colonies from each of the three clusters in **a** and control ("untreated") colonies on antibiotic-free agar. Each solid line in **b** represents the mean growth curve profile of colonies within each cluster (untreated and treated + clustered). In **c**, box plots indicate the median (center line), 25th and 75th percentiles (bounds of box), and whiskers extending to ±1.5× the interquartile range (IQR); points beyond the whiskers are shown as outliers. Violin plots represent the distribution and density of the data. Fold change in IC$_{50}$ of INH (**d**) and relative fitness (**e**) for isolates in each of the three clusters. In **d** and **e**, box plots indicate the median (center line),

25th and 75th percentiles (bounds of box), whiskers extending to ±1.5× the IQR; overlayed by the mean value for each isolate (points). Statistical significance was evaluated using the two-sided Student's t-test and $p$ values were Bonferroni corrected for multiple testing (ns = not significant, adj. $p \geq 0.05$). **f** Fold change in IC$_{50}$ of INH and fitness in antibiotic-free growth medium for at least 12 isolates from each cluster ($n = 55$) relative to untreated controls across four replicates (from duplicate cultures across two independent experiments). Error bars indicate the mean ± SEM. The gray box centered on coordinates (1, 1) represents standard errors in relative fold changes in INH IC$_{50}$ (width of box) and fitness (height of box) across all untreated control isolates ($n = 12$). **g** Time-kill curves with 100× minimum inhibitory concentration (MIC) INH (1 mg/mL) for each of the three mutants and the wildtype strain. Solid lines represent average kill curve for 12 replicates of each strain. Source data are provided as a Source Data file.

summary, the genotypic and phenotypic analyses revealed that while 85% of the 40 isolates may have survived low-dose INH treatment through induction of intrinsic mechanisms of phenotypic tolerance or resistance, point mutations in three genes (*ohrR*, *mfs1*, and *ntaA_5*) conferred low-level resistance, without fitness tradeoff, across 15% of the remainder isolates. Since mutations in these three genes also conferred high-level tolerance to INH, here onwards we will call these LLRT mutants.

## LLRT mutations potentiate rapid gain of high-level INH resistance with no fitness tradeoff

High-level INH$^R$-conferring mutations are also associated with significant fitness tradeoff and, therefore, unlikely to survive in the absence of antibiotic within a naïve wildtype mycobacterial population[10,43]. We performed the fluctuation test to investigate whether the altered fitness landscape of LLRT mutants, which we have demonstrated can co-exist in the absence of antibiotic within a naïve

**Table 1 | Mutations in three genes confer low-level resistance and tolerance to INH**

| Isolate | Mutated Gene | Allele characterization | | Mtb ortholog | INH susceptibility (IC$_{50}$ fold change) | Fitness relative to wildtype | INH MDK$_{99}$ |
|---|---|---|---|---|---|---|---|
| | | Mutation | Impact | | | | |
| G8 | *ohrR* MSMEG_0448 | g12ins | P4* Frameshift | *oxyR'*, Rv2427A hydrogen peroxide-inducible genes activator (orthologous pathway) | Low-level resistance (3.09 ± 1.07) | 0.992 ± 0.023 | Tolerant |
| G16 | *ohrR* MSMEG_0448 | g12ins | P4* Frameshift | *oxyR'*, Rv2427A hydrogen peroxide-inducible genes activator (orthologous pathway) | Low-level resistance (2.61 ± 0.588) | 1.058 ± 0.009 | Tolerant |
| B12 | *mfs1* MSMEG_2380 | c314t | G105D | Rv2994 major facilitator superfamily transporter Frac. Identity: 0.597, Coverage: 96% | Low-level resistance (1.39 ± 0.257) | 0.919 ± 0.073 | Tolerant |
| H4 | *ntaA_5* MSMEG_6641 | g149del | E50* Frameshift | none | Low-level resistance (1.38 ± 0.122) | 0.987 ± 0.029 | Tolerant |
| H8 | *ntaA_5* MSMEG_6641 | g149del | E50* Frameshift | none | Low-level resistance (1.35 ± 0.061) | 1.032 ± 0.037 | Tolerant |
| B103 | *fas1* MSMEG_4757 | t530c | E177G | *fas1*, Rv2524c fatty acid synthase 1Frac. Identity: 0.816, Coverage: 99% | Susceptible (0.975 ± 0.104) | 1.006 ± 0.033 | ND |

Isolate(s): Notation is evolutionary line (letter) and colony (number).

Mutated gene: *ohrR* organic hydroperoxide reductase regulator, *mfs1* major facilitator superfamily, *ntaA_5* xenobiotic compound monooxygenase, *fas1* fatty acid synthase 1.

Allele characterization: ins, insertion; del, deletion.

Mtb ortholog: Reciprocal best hit between Msm mc²155 and Mtb H37Rv found with BLASTP[129] and MMseqs2[130].

INH IC$_{50}$ fold change: Isolate INH IC$_{50}$ fold change with respect to the parental wildtype strain.

Fitness WRT wildtype: Isolate uninhibited growth (area under growth curve) with respect to parental strain.

INH MDK$_{99}$: Isolate INH tolerance as determined by time kill assay.

population, could also potentiate spontaneous gain of high-level INH$^R$ [44,45]. Twelve lines of the *fas1::E177G* mutant, each of the three LLRT mutants described above and wildtype Msm were inoculated at low cell densities (~200 cells in 200 μl of 7H9 broth) and grown in the absence of antibiotic to mid-log phase. Each culture was plated on 7H10 agar with and without 50× MIC (500 μg/mL) INH and the number of INH$^R$ mutant colonies and total population size were estimated with CFU counting. The FALCOR tool[44] was then used to estimate the rate of gain of INH$^R$ using two independent methods, frequency and Ma-Sandri-Sarkar Maximum Likelihood Estimator (MSS-MLE)[44]. The fluctuation test results showed that all three LLRT mutants acquired high-level INH$^R$ at up to 6-fold higher rate, relative to the wildtype strain (Fig. 2a, b).

Next, we measured IC$_{50}$ of INH in LLRT-derived INH$^R$ mutants, and investigated whether the gain of resistance in these strain backgrounds was associated with fitness tradeoff in the absence of antibiotic. Dose response assays demonstrated that the level of INH$^R$ acquired in LLRT strain backgrounds was comparable (*mfs1::G105D* and *ntaA_5::E50*fs*) or significantly higher (*ohrR::P4*fs*) relative to level of INH$^R$ acquired in the wildtype background (Fig. 2c). Furthermore, there was high variability in the level of INH$^R$ across wildtype-derived INH$^R$ mutants, suggesting a broad spectrum of evolved resistance mechanisms. By contrast, relative to INH$^R$ strains derived from the wildtype background, significantly higher levels of INH$^R$ were gained consistently (i.e., with low variability in IC$_{50}$ measurements across isolates) in the *ohrR::P4*fs* background (Bonferroni corrected $p$ value = 6.87×10$^{-7}$, Fig. 2c). More importantly, while gain of INH$^R$ was associated with significant fitness tradeoff in the wildtype, *mfs1::G105D*, and *ntaA_5::E50*fs* strain backgrounds, acquisition of INH$^R$ in the *ohrR::P4*fs* strain background was associated with minimal fitness cost (Fig. 2d). Thus, findings from the dose response and fitness assays demonstrated that LLRT mutations occur at higher frequency within a naïve population, and are enriched rapidly by brief exposure to low-dose antibiotic treatment. Further, these findings provide evidence that LLRT mutations are likely pre-resistance mutations that potentiate, with minimal fitness tradeoff, rapid gain of high-level INH$^R$ in *Mycobacterium spp.*

**The *ohrR::P4*fs* mutation alleviates oxidative stress and constrains the evolutionary trajectory for acquiring high-level INH$^R$**

To uncover the mechanisms by which high-level of INH$^R$ was acquired in the wildtype- and *ohrR::P4*fs* strain backgrounds, we sequenced the genomes of 12 small (s) and 9 large (L) INH$^R$ colonies from the wildtype background and 12 small (s) and 12 large (L) INH$^R$ colonies from the *ohrR::P4*fs* background. We did not see any distinguishable characteristics between small and large colonies in terms of acquired mutations, INH IC$_{50}$, or fitness. However, while 14/21 (66%) isolates from the wildtype background had acquired nonsynonymous point mutations in *ndh* (which encodes a NADH dehydrogenase), 24/24 (100%) isolates from the *ohrR::P4*fs* background had acquired nonsynonymous frameshift mutations in the mycothiol biosynthesis (*msh*) operon genes (Fig. 3a). By contrast, 0/21 isolates from the wildtype background harbored *msh* mutations, and 0/24 isolates from the *ohrR::P4*fs* background harbored *ndh* mutations. Interestingly, none of the isolates had gained mutations in canonical INH resistance-conferring loci aside from *ndh*. For a full list of mutations identified in the sequenced isolates, see the Source Data file. Phenotypic characterization of each of these isolates revealed that gain of high-levels of INH$^R$ in the wildtype background was associated with significant fitness loss, where only 3/21 wildtype-derived INH$^R$ mutants had at least 90% fitness relative to the wildtype itself in the absence of antibiotic treatment. By contrast, the majority (17/24) of the high-level INH$^R$ mutants derived from the *ohrR::P4*fs* background had at least 90% fitness relative to the wildtype strain in the absence of antibiotic treatment (Fig. 3c). These results suggest that the mechanism by which

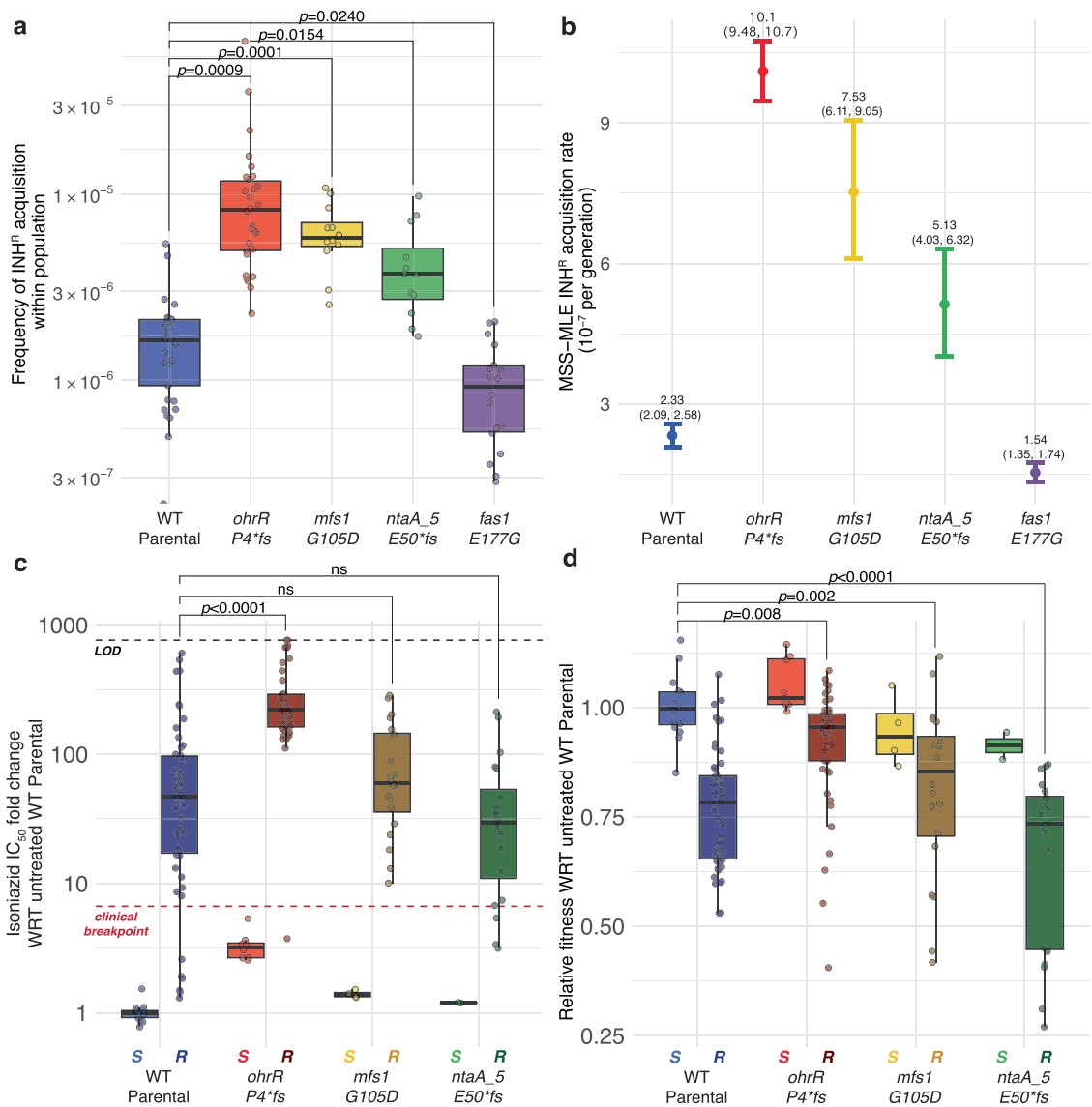

**Fig. 2 | LLRT strains acquired high-level INH^R-conferring mutations at a high rate, with minimal fitness tradeoff.** Fluctuation assay results using the frequency method (**a**) or the Ma-Sandri-Sarkar Maximum Likelihood Estimator method (**b**). Points and error bars in **b** represent the median and 95% confidence interval (CI), respectively. The median and upper and lower 95% CI bounds are labeled for each strain. Results are representative of two independent experiments. Fold change in INH IC$_{50}$ (**c**) and relative fitness (**d**) of INH^R mutants (labeled 'R') and the parental strains (labeled 'S') from which they were derived. Relative fitness was calculated as the ratio of the AUC of each isolate with respect to the average AUC of susceptible wildtype controls after 48 hours of growth in the absence of INH. The black

horizontal dashed line in **c** represents the upper limit of detection (LOD) for resistance, indicating ≥ 50% AUC at the maximum concentration tested relative to the strain's untreated AUC. The red horizontal line represents the clinical break-point for high-level INH resistance ( ≥ 6.6× fold change in IC$_{50}$). In **a**, **b**, and **c**, box plots indicate the median (center line), 25th and 75th percentiles (bounds of box), and whiskers extending to ±1.5× the IQR; overlayed by the mean value for each biological replicate of each isolate (points). Statistical significance was evaluated using the two-sided Welch's t-test and all *p* values are Bonferroni corrected for multiple testing (ns = not significant, adj. *p* ≥ 0.05). Source data are provided as a Source Data file.

*ohrR::P4*fs* mutation buffers the fitness cost of resistance, also constrains the evolutionary trajectory (i.e., mutations in mycothiol biosynthesis genes) through which high-level INH^R is acquired in this specific genomic background. For the phenotypic characterization of all fluctuation assay-derived isolates from wildtype and LLRT backgrounds, see Supplementary Fig. 3.

Based on literature[46] and our own findings reported here, we hypothesized that disruption of mycothiol biosynthesis by itself may confer much lower level of resistance to INH, and would carry a substantial fitness trade-off. To test this hypothesis, we performed dose response assays on an Msm strain with a single gene *in frame* deletion of *mshA*, and determined that while the Δ*mshA* strain had ~22× higher IC$_{50}$ relative to its parental strain (*p* value = 0.0473), the *ohrR-mshA/C* double

mutants generated in this study consistently showed > 200× higher IC$_{50}$ for INH (Fig. 3b–c). Furthermore, the increased INH^R of the Δ*mshA* single gene mutant was associated with a significant fitness tradeoff (~ 48% AUC of wildtype) (Fig. 3c), which is consistent with a prior report that MSH mutants have poor fitness due to increased oxidative stress[46].

### The *ohrR::P4*fs* mutation potentiates acquisition of high-level INH^R through the alleviation of oxidative stress

Findings from the genome re-sequencing analysis suggested that the wildtype and *ohrR::P4*fs* strains acquired high-level INH resistance through distinct mechanisms associated with mutations in the *ndh* and *mshA* genes, respectively. Disruptive mutations in *ndh* are known to confer high-level INH resistance, both in Msm and in Mtb[40,46–51], by

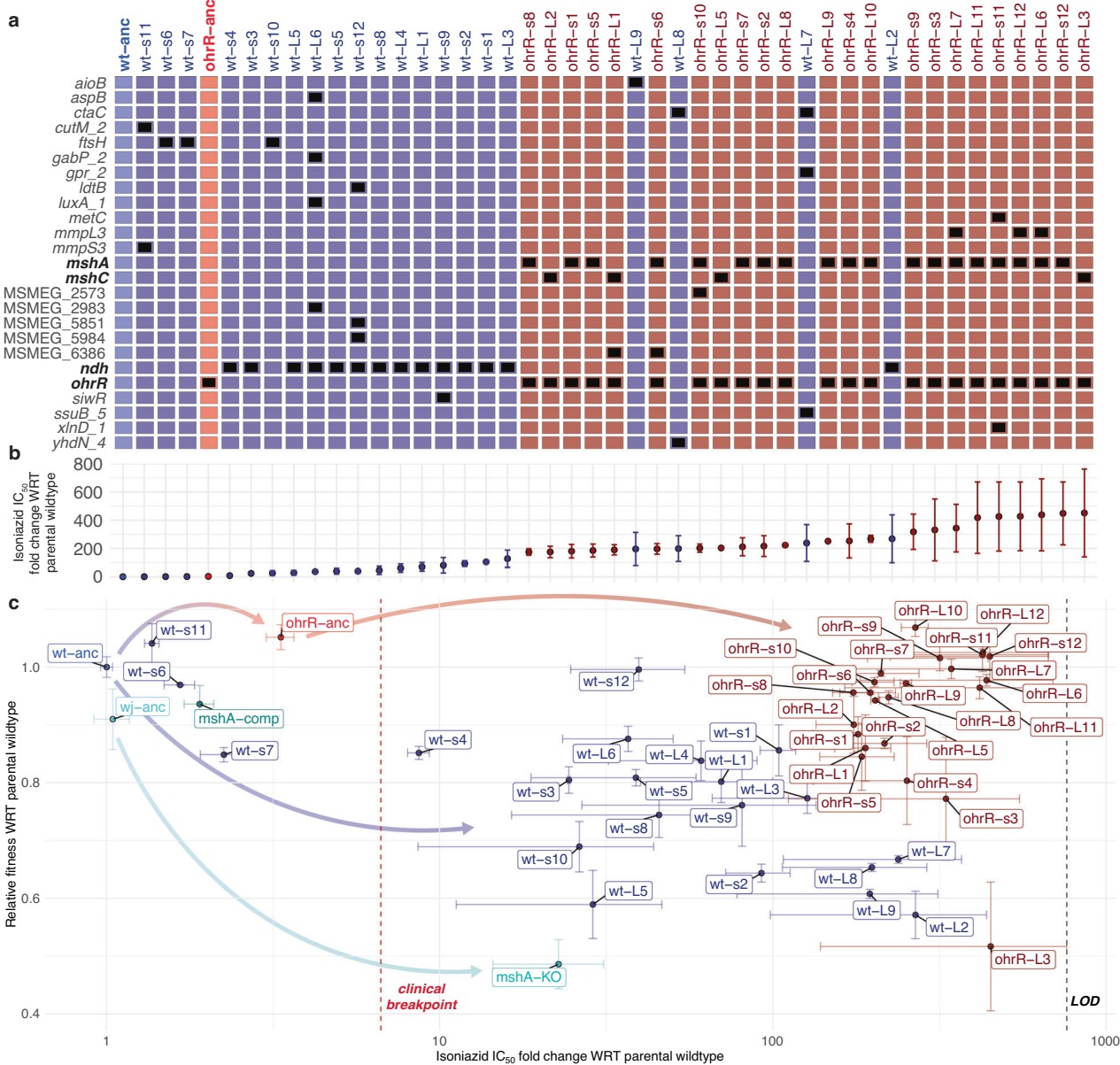

**Fig. 3 | Loss-of-function mutations in *ohrR* potentiate the subsequent gain of high-level INH^R through a constrained evolutionary trajectory. a** Grid plot of all sequenced isolates from wildtype and *ohrR::P4*fs* backgrounds, rank-ordered by their INH IC$_{50}$. Black boxes indicate nonsynonymous mutations at specific loci (rows) in each strain (columns) identified by WGS analysis. For a full list of specific mutations, see the Source Data file. Specific genes of interest are bolded for emphasis. **b** Corresponding fold change in IC$_{50}$ of INH of each isolate in **a**. Error bars represent the standard error across two replicates in two independent experiments ($n = 4$). **c** Scatter plot of relative fold changes in INH IC$_{50}$ (x-axis) and fitness (y axis) in the absence of antibiotic of all INH^R isolates derived from wildtype and *ohrR::P4*fs* backgrounds. Relative change in IC$_{50}$ (quantified using dose-response assays) is with respect to the average IC$_{50}$ of the ancestral wildtype strain from which the isolates were derived. Relative fitness was calculated as the ratio of the AUCs of growth curves of each isolate and the wildtype parental strain from which

they were derived, after 48 hours of growth in the absence of INH. Error bars represent the standard error across two replicates in two independent experiments ($n = 4$). The black horizontal dashed line in **c** represents the upper limit of detection (LOD) for resistance. The red horizontal line represents the clinical breakpoint for high-level INH resistance ($\geq 6.6\times$ fold change in IC$_{50}$). The colored arrows illustrate the evolutionary trajectory across the resistance and fitness landscape of various genetic backgrounds upon gaining high-level INH^R. Strain labels: wt-anc: Nitin Baliga Lab wildtype Msm mc$^2$155; wj-anc: William Jacobs Lab wildtype Msm mc$^2$155; *mshA*-KO: Δ*mshA* knockout derived from W.J. wildtype; *mshA*-comp: Δ*mshA* pMV361::*mshA* complemented strain derived from W.J. wildtype; mfs1-anc: *mfs1::G105D*; ntaA-anc: *ntaA_5::E50*fs*; ohrR-anc: *ohrR::P4*fs*. All remaining strains were derived from fluctuation assay of wildtype or *ohrR::P4*fs* backgrounds. Source data are provided as a Source Data file.

increasing the NADH pool, which overrides competitive inhibition of InhA by the INH-NADH adduct[52,53]. Indeed, *ndh* mutants selected in the wildtype strain background had significantly higher NADH/NAD+ ratio with or without $1\times$ IC$_{50}$ INH treatment (Fig. 4a and Supplementary Fig. 2). As expected, there was no change in NADH/NAD+ ratio in the *ohrR* or *mshA* single or double mutants, confirming a different underlying mechanism of INH^R in these strains.

Mutations in mycothiol (MSH) biosynthesis genes, such as *mshA*, *mshB*, and *mshC*, have been associated with varying levels of INH resistance across *Mycobacterium* species, but the observed phenotypes are inconsistent and appear to depend on the broader genomic context[46,51,54]. Studies in Msm and Mtb have reported a wide range of INH resistance levels, from low-level (~2× MIC) to very high-level (>100× MIC), depending on the specific mutation and strain

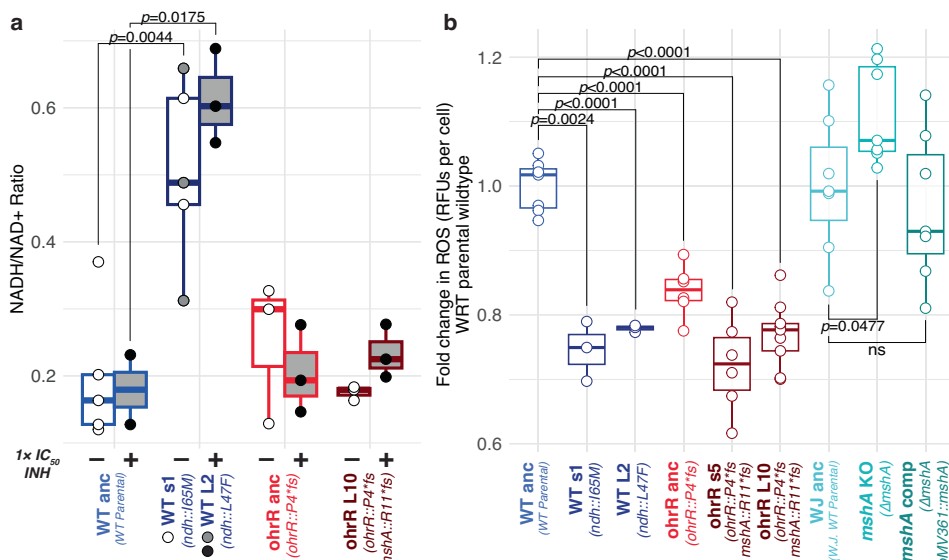

**Fig. 4 | Mutations that ameliorate oxidative stress precede and potentiate the gain of subsequent high-level INHR mutations. a** NADH/NAD+ ratio during log phase of the wildtype parental, *ohrR::P4*fs* mutant, and INHR strains evolved from each background, in the absence (indicated with "(−)") and presence (indicated with "(+)") of 1× $IC_{50}$ INH. Due to sample loss, INH-treated WT anc and untreated WT s1 are represented by two biological replicates. Due to the similar genetic background between WT s1 and WT L2, untreated data for these strains were grouped together. **b** Endogenous ROS levels were measured using H2DCFDA in log phase cultures. Fold change in relative fluorescent units (RFUs) per cell was determined by comparing the RFUs per cell of each strain to the parental wildtype strain from which they were derived, across at least three biological replicates. Strains are labeled with the name of the background from which they were derived and their genotype relative to the wildtype parental strain. Point mutants are named by their colony size (s for small, L for large) and the isolate number. In (**a**) and (**b**), box plots indicate the median (center line), 25 and 75th percentiles (bounds of box), and whiskers extending to ±1.5× the IQR; overlayed by the measured value for each biological replicate (points). Significance was evaluated using the two-sided Welch's t-test (ns = not significant, $p \geq 0.05$). Source data are provided as a Source Data file.

background. In Msm, early work identified mutants deficient in MSH that were hypersensitive to rifampicin but highly resistant to INH, suggesting that MSH participates in INH activation, possibly by maintaining KatG or InhA in a reduced, reactive state[55]. More specifically, *mshA* and *mshC* knockouts in Msm exhibited moderate INH resistance (4-8× MIC) alongside hypersensitivity to oxidative stress[56,57]. Similarly, *mshD::Tn5* mutants, which accumulate the thiol precursor Cys-GlcN-Ins instead of MSH, showed extreme resistance to INH (MIC>100 µg/mL), indicating that MSH itself is required for INH activation[58]. It was also reported that transposon mutants of *mshA* (*mshA::Tn5*) exhibited very high INH resistance (>250× MIC)[59], and that increased oxidative stress due to lack of mycothiol in these mutants was compensated by overexpression of *ohr* and ergothioneine[60,61].

Together, these findings suggest that while blocking prodrug activation in the Δ*mshA* mutant may lessen INH-induced oxidative stress[62,63], loss of MSH increases endogenously generated oxidative stress. If confirmed, this would explain why gain of resistance to INH is less likely to occur in a wildtype background through selection of fitness-compromising loss-of-function *mshA* mutations. Independently, it has been demonstrated that knocking out *ohrR* improves the oxidative stress response of Msm through constitutive overexpression of *ohr*[24,64], providing a plausible mechanism by which the *ohrR::P4*fs* pre-resistance mutation may have potentiated the gain of INHR through selection of loss-of-function mutations in *mshA*. To test this hypothesis, we measured endogenous ROS levels with or without INH treatment during log-phase growth of two *ndh* mutants (*ndh::*I65M, *ndh::*L47F), two *ohrR::P4*fs - mshA::R11*fs* double mutants, Δ*mshA* and its complemented strain, as well as their parental strains. Not surprisingly, ROS levels were relatively low in the *ndh* mutants, wherein the fitness tradeoff was due to disrupted redox balance. As expected, we observed higher ROS levels in the Δ*mshA* strain, which were restored to the wildtype levels in the complemented strain. ROS levels were significantly lower in the *ohrR::P4*fs* pre-resistant mutant, and,

strikingly, even lower in the INHR *ohrR::P4*fs - mshA::R11*fs* double mutants (Fig. 4b, Supplementary Fig. 4).

## Brief exposure to growth sub-inhibitory oxidative stress potentiates evolution of INHR

Our findings suggest that MSH-deficiency-mediated INHR is unlikely to emerge in a naïve genomic background, requiring the fitness cost to be preemptively buffered by disruption of OhrR-mediated repression of Ohr. Indeed, disruption of OhrR in the *ohrR::P4*fs* strain accelerated the emergence of high-level INHR by ~6-fold through loss-of-function *mshA* mutations (Fig. 5). Together these findings demonstrate that preemptive mitigation of oxidative stress facilitates the selection of INHR mutations. Building on this, we hypothesized that the activation of the OSR through pre-exposure to low-level oxidative stress should also potentiate the evolution of INHR. To test this hypothesis, we performed a fluctuation assay on wildtype Msm with and without pretreatment with a growth sub-inhibitory dose (1× $IC_{50}$; 135 ± 7.6 µM) of cumene hydroperoxide (CHx) (Supplementary Fig. 5) for 24 h prior to plating out on 7H10 agar plates with 50× MIC INH. Indeed, pretreatment with 1× $IC_{50}$ CHx for 24 hours increased the rate of gain of high-level INHR by up to 2.7-fold relative to wildtype Msm not challenged with CHx ($p$ value = 4.51×10$^{-9}$; Fig. 5e, f). Taken together, our results show that activation of OSR either by genetic perturbations (e.g., disruption of *ohrR*) or pre-exposure to oxidative stress can accelerate the emergence of INHR in Msm.

## Bayesian analysis associates mutations in OSR genes to the potentiation and spread of strains with clinical resistance to multiple antibiotics

We investigated whether oxidative stress-ameliorating mutations may have also potentiated the emergence of INHR in clinical strains of Mtb through the analysis of whole genome sequencing (WGS) data for 1578 clinical Mtb samples from Ho Chi Minh City, Vietnam, with drug

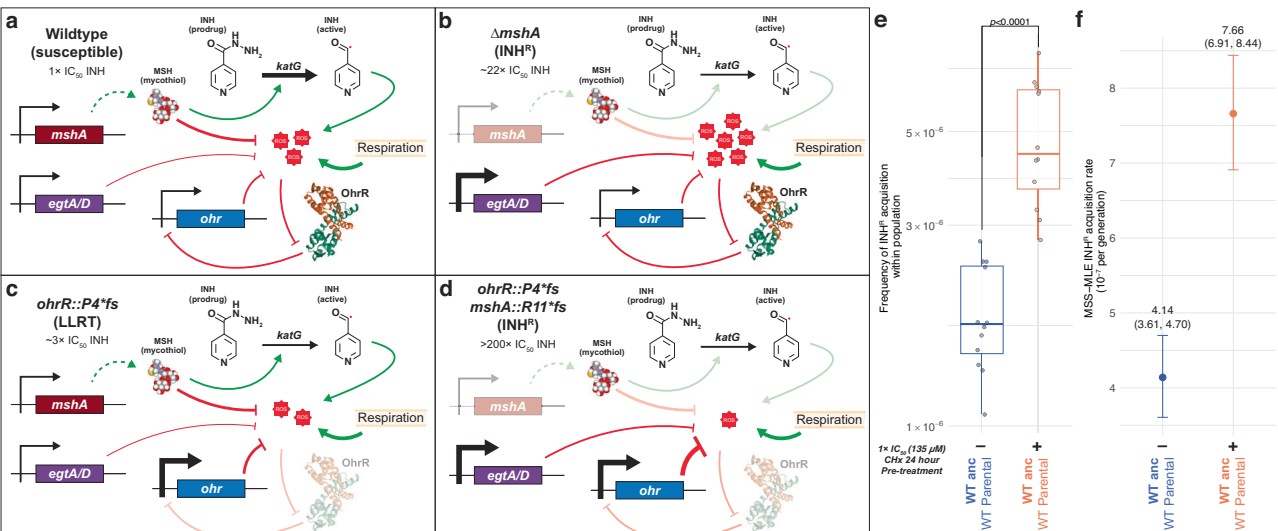

**Fig. 5 | Proposed mechanism by which a loss-of-function mutation in *ohrR* supports disruptive mutations in mycothiol biosynthesis pathway genes to potentiate the evolution of INH^R in Msm. a** Genes of the *msh* operon encode biosynthesis of mycothiol (MSH). While the primary role of MSH is in mitigating oxidative stress (ROS) generated by respiration, MSH also promotes KatG-mediated activation of INH[26,57]. Increase in ROS, due to INH treatment, triggers *ohr* expression by peroxide-mediated dissociation of the repressor OhrR from its promoter[24,64,131]. The increased expression of Ohr helps maintain physiologically tolerable ROS levels. **b** While loss of MSH in the wildtype background confers elevated resistance to INH ( ~ 22× $IC_{50}$), it also results in significant increase in ROS levels, which leads to growth inhibition (loss of fitness). **c** Loss-of-function mutations in *ohrR*, such as *ohrR::P4*fs*, derepress *ohr* expression[25,61,64] resulting in significantly lower ROS levels and LLRT to INH ( ~ 3× $IC_{50}$). **d** The lower ROS environment in the *ohrR::P4*fs* background supports depletion of MSH through subsequent selection of loss-of-function mutations in *msh* operon genes,

preventing INH activation and resulting in high-level INH^R phenotype (>200× $IC_{50}$). The loss of the protective effect of MSH is compensated by the elevated Ohr levels[46,51,52,61]. The protein model of OhrR is an orthologous structure from *Xanthamonas campestris* (PDB 2PFB)[132]. The thickness of the edges represents the expected change in expression or activity of the given interaction. Fluctuation assay results to estimate rate of acquisition of INH^R after (±) pre-treatment with 1× $IC_{50}$ cumene hydroperoxide (CHx) (135 μM) across 12 biological replicates, using the frequency method (**e**) or the Ma-Sandri-Sarkar Maximum Likelihood Estimator method (**f**). In **e**, box plots indicate the median (center line), 25 and 75th percentiles (bounds of box), and whiskers extending to ±1.5× the IQR; with points showing the mean value across three technical replicates for each biological replicate ($n = 12$). Points and error bars in f represent the median and 95% CI, respectively. The median and upper and lower 95% CI bounds are labeled for each strain. Statistical significance was evaluated using the two-sided Student's t-test. Source data are provided as a Source Data file.

susceptibility testing (DST) results for a spectrum of frontline anti-TB antibiotics[65] (Fig. 6a–b). Briefly, WGS reads were aligned to a reconstructed ancestral Mtb genome[66] and analyzed with a custom variant calling pipeline adapted from[67] (https://github.com/baliga-lab/bwa_pipeline) to identify 1,140,577 fixed and 194,263 unfixed (at least 10% alternative allele frequency (AAF)) mutations (316,823 intergenic and 1,018,017 nonsynonymous protein-coding mutations) across 5233 genomic loci within each sample (Fig. 6c). We then applied a Bayesian probability (BP) framework with permutation testing to the genotypic and phenotypic profiles of the Mtb samples to identify specific loci that were significantly associated with acquisition of INH resistance (Fig. 6d) (Methods). Of the 5233 intergenic and genic loci, mutations in 393 loci had significant BP (FDR-corrected $p$ value ≤ 0.05), of which 216 (54.96%) were protein-coding genes. As expected, protein-coding genes in loci with significant Bayes probabilities included canonical mechanisms of INH resistance, including *katG* (BP = 0.858), *inhA* (BP = 0.588), *oxyR'-ahpC* (BP = 0.667), and *katG-furA* (BP = 0.8). Significant Bayes probabilities of *pknH-embR* (BP = 0.714), *pyrR* (BP = 0.897), *pncA* (BP = 0.876), and *rpoB* (BP = 0.737), which are associated with canonical mechanisms of resistance to other frontline TB drugs, such as ethambutol, pyrazinamide and rifampicin, recapitulated the phenomenon of cross-resistance[68] and evolutionary dependency[66], whereby resistance to one drug (in particular INH) increases the risk of treatment failure and acquiring multi-drug resistance in TB[69,70]. Interestingly, relative to their background proportion in the overall dataset (1614/5233, 30.84%), intergenic loci were overrepresented amongst loci with significant Bayes probabilities (177/393, 45.04%; $p$ value = 5.12×10^{-10}). This finding suggested that many non-canonical

regulatory mutations may have played a significant role in the emergence of INH^R clinical strains of Mtb.

Using results from a genome-wide fitness screen, we then discovered that the 207/216 genes included in the screen with significant Bayes probabilities were significantly overrepresented (hypergeometric test $p$ value = 0.0261) among CRISPRi knockdown strains that were depleted (19/207 genes) or enriched (9/207 genes, FDR ≤ 0.01, absolute value log₂ fold change ≥ 1) after 5 days of high-dose INH treatment[71] (Fig. 6e). Given the interesting finding that the Bayesian analysis had uncovered pre-resistance mutations that may have potentiated multidrug resistance, we also analyzed the CRISPRi screening results for evidence supporting the functional association of high BP genes with resistance to other antibiotics. This analysis demonstrated that although the Bayesian analysis was performed in the context of INH^R, CRISPRi strains targeting high BP genes were significantly enriched or depleted by treatment with multiple other antibiotics, including rifampicin, ethambutol, clarithromycin, streptomycin, and vancomycin (Table 2, Supplementary Data 2).

Notably, CRISPRi knockdown strains of *mshA*, *mshB*, and *mshC* were among the most enriched during treatment with INH, providing additional evidence for fitness advantage of disrupting mycothiol biosynthesis in Mtb during INH treatment. The high Bayes probability of the *oxyR'-ahpC* intergenic region as well as the finding that the CRISPRi knockdown strain of *ahpC* was among the most depleted after 5 days of high-dose INH treatment, underscored the importance of oxidative stress mitigation of AhpC function, the Mtb ortholog of Ohr, and its regulation in the emergence of INH resistance. Given that Mtb routinely experiences oxidative stress inside the host during infection,

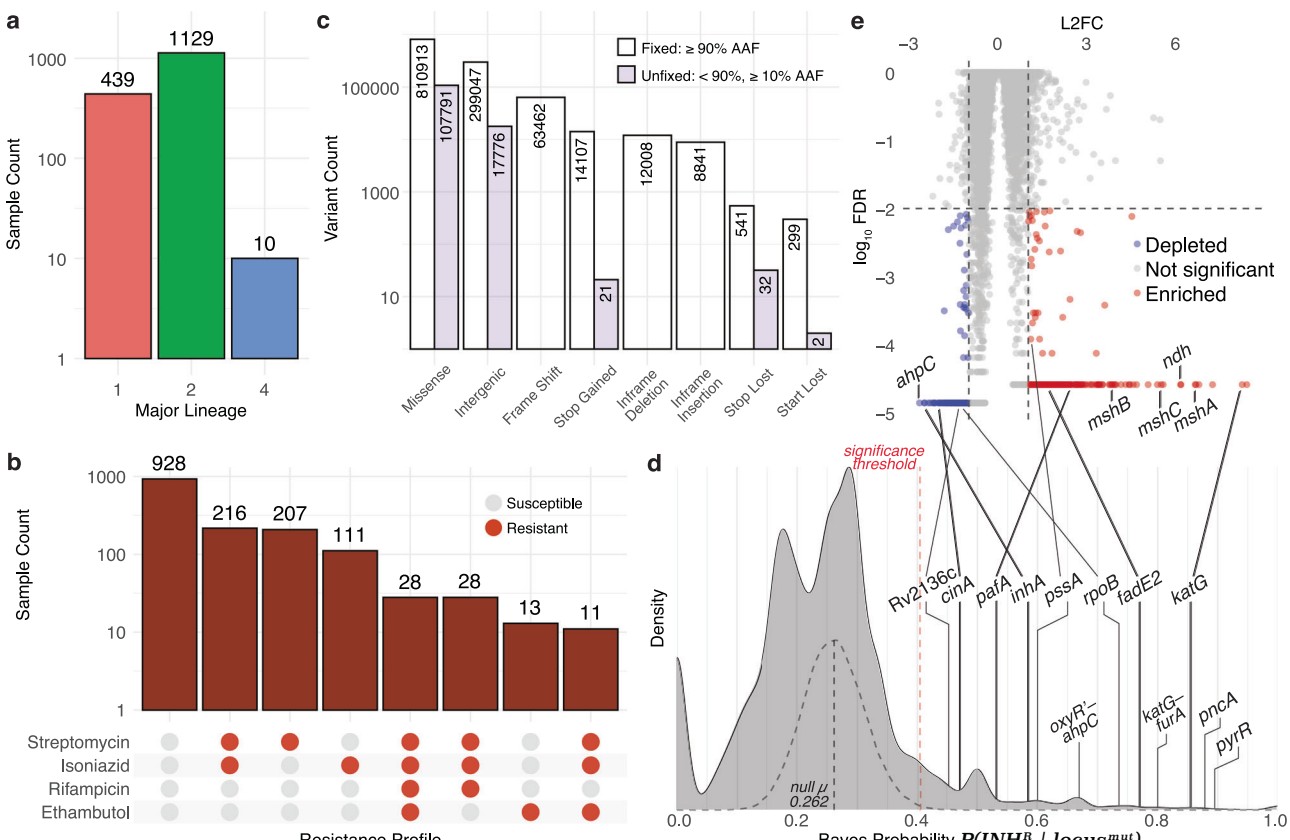

**Fig. 6 | A Bayesian framework to identify genomic signatures of pre-resistance in clinical Mtb isolates. a** Counts of all isolates from each major lineage, as determined by TB-Profiler[124]. **b** Counts of all isolates with their corresponding susceptibility to four frontline TB drugs, as determined by drug susceptibility testing (DST). **c** Counts of all types of variants called across the entire dataset after removing synonymous variants from the analysis as well as variants occurring in less than 1% or more than 95% of all isolates. Clear boxes represent fixed variants ($\geq$ 90% AAF) and purple boxes represent unfixed variants (< 90%, $\geq$ 10% AAF). **d** Bayes probability distribution for all loci (genic and intergenic) throughout the Mtb genome for the entire dataset. The null Bayes probability distribution after > 5000 randomized permutations is overlayed as a gray dashed density plot. The

significance threshold in the dashed red line represents the threshold value above which Bayes probabilities have an FDR-corrected $p$ value $\leq$ 0.05 based relative to the null distribution. Notable loci with significant Bayes probabilities are labeled on the density plot (FDR-corrected $p$ value $\leq$ 0.05). **e** Log$_2$ fold change (L2FC) and false discovery rates (log$_{10}$ FDR-corrected $p$ value) for each gene in the CRISPRi library after 5 days of high-dose INH treatment[71]. This figure is a modified version of what can be found in previous literature[71]. (**d, e**) Notable loci are labeled as examples of genes with significant Bayes probability for association with INH$^R$ and whose corresponding CRISPRi knockdown strains were enriched or depleted during high-dose INH treatment[71]. Source data are provided as a Source Data file.

we further explored whether mutations in genes generally involved in the mitigation of oxidative stress may have potentiated the emergence of INH resistance. In order to do this, we first compiled a list of 68 genes associated with mitigation of oxidative stress in Mtb by leveraging assigned gene ontology (GO) terms (antioxidant activity, GO:0016209; cellular oxidant detoxification, GO:0098869; removal of superoxide radicals, GO:0019430; mycothiol-dependent detoxification, GO:0010127; response to oxidative stress, GO:0006979; response to nitrosative stress, GO:0051409). Then, we constructed a regulatory network for these genes by leveraging two independent datasets: the transcription factor over-expression (TFOE) dataset that catalogs sets of genes that are differentially expressed as a consequence of overexpressing any of 209 transcription factors (TFs) in the Mtb genome[72] and experimentally-mapped binding sites for 143 TFs through chromatin immunoprecipitation sequencing (ChIP-seq)[73,74] (Methods). The final 159 gene OSR network constructed using GO, ChIP-seq and TFOE, includes all known GO-annotated OSR genes (58 genes and 10 TF regulators), their own 27 transcriptional regulators, as well an additional 64 downstream target genes (Fig. 7). Strikingly, the final OSR network of Mtb (122 genes regulated by 37 TFs) was enriched for 19 genes and 9 intergenic regions with significant BP for potentiating INH resistance (hypergeometric test $p$ value = 1.09×10$^{-7}$). Many genes within the OSR network that also have

significant BP have been previously implicated in mono- or multi-drug resistance, including Rv2136c[75], *pknH-embR*[76-78], *whiB3*[79], *katG-furA*[80-83], Rv0238[84], and *sigH*[85,86] (Supplementary Data 1). Upon further examination of overlap with additional CRISPRi screens, we discovered that the OSR network genes were also significantly associated with Mtb survival in the presence of multiple other antibiotics, including bedaquiline, clarithromycin, linezolid, and streptomycin (Table 2). For a full list of genes identified in this analysis, see Supplementary Data 2. Thus, findings from our studies on Msm, the Bayesian analysis of clinical strains of Mtb, CRISPRi knockdown screens[71], and the analysis of the OSR network of Mtb, together demonstrated that mutations that mitigate oxidative stress may have played an important role in the emergence of multi-drug resistance in Mtb.

## Discussion

The findings presented in this study reveal that oxidative stress management and exposure to low-dose antibiotic may play a critical and previously underappreciated role in the evolution of antibiotic resistance in Mtb. The data support a model wherein LLRT mutations that enhance oxidative stress resilience, such as those affecting the OxyR-AhpC regulatory axis, create a permissive genomic background that accelerates the acquisition of high-level drug resistance without incurring prohibitive fitness costs.

**Table 2 | Overrepresentation of genes with significant BP or OSR network genes among CRISPRi knockdown strains during antibiotic treatment**

| CRISPRi treatment | | Gene Group | BP vs. CRISPRi hypergeometric test *p* value | n Sig. BP genes | OSR vs. CRISPRi hypergeometric test *p* value | n OSR genes |
|---|---|---|---|---|---|---|
| **Antibiotic** | **Dose / Day(s)** | | | | | |
| INH | Medium / 5 | Depleted | 0.045011 | 2 | NA | 0 |
| | High / 5 | Enriched | NA | 0 | 0.019481 | 14 |
| | | Depleted | 0.001180 | 19 | 0.008385 | 13 |
| | | Responder | 0.026146 | 28 | 0.000421 | 27 |
| | High / 10 | Enriched | NA | 0 | 0.025310 | 12 |
| | | Depleted | 0.017955 | 16 | 0.046859 | 10 |
| | | Responder | NA | 0 | 0.003141 | 22 |
| EMB | High / 5 | Enriched | 0.008613 | 27 | NA | 0 |
| | | Responder | 0.005087 | 35 | NA | 0 |
| | High / 10 | Enriched | 0.046073 | 15 | NA | 0 |
| | | Responder | 0.026895 | 20 | NA | 0 |
| CLR | Low / 5 | Enriched | 0.008638 | 6 | 0.010529 | 5 |
| | | Responder | NA | 0 | 0.009757 | 9 |
| | Low / 10 | Enriched | 0.000131 | 6 | 0.026120 | 3 |
| | | Depleted | NA | 0 | 0.004428 | 9 |
| | | Responder | 0.016457 | 11 | 0.000413 | 12 |
| | Medium / 5 | Enriched | 0.025796 | 6 | 0.006929 | 6 |
| | | Responder | NA | 0 | 0.001852 | 11 |
| | Medium / 10 | Enriched | 0.002800 | 6 | NA | 0 |
| | | Depleted | NA | 0 | 0.039083 | 8 |
| | | Responder | 0.016457 | 13 | 0.010002 | 11 |
| | High / 5 | Enriched | 0.004981 | 10 | NA | 0 |
| | | Responder | NA | 0 | 0.012131 | 11 |
| | High / 10 | Enriched | 0.003821 | 7 | NA | 0 |
| | | Responder | 0.016638 | 15 | NA | 0 |
| RIF | Medium / 1 | Depleted | 0.046011 | 7 | NA | 0 |
| | High / 10 | Enriched | 0.019487 | 2 | NA | 0 |
| STR | High / 1 | Enriched | 0.030564 | 3 | NA | 0 |
| | | Depleted | NA | 0 | 0.018035 | 2 |
| | | Responder | NA | 0 | 0.023490 | 3 |
| | High / 5 | Enriched | 0.005130 | 6 | NA | 0 |
| | | Depleted | NA | 0 | 0.030776 | 3 |
| | | Responder | 0.048033 | 6 | 0.036504 | 5 |
| BDQ | Medium / 10 | Enriched | NA | 0 | 0.004958 | 2 |
| LZD | Medium / 1 | Enriched | NA | 0 | 0.040363 | 2 |
| | Medium / 10 | Enriched | NA | 0 | 0.015642 | 2 |
| | High / 5 | Depleted | NA | 0 | 0.008712 | 6 |
| | | Responder | NA | 0 | 0.043427 | 11 |
| | High / 10 | Responder | NA | 0 | 0.017540 | 6 |
| VAN | Medium / 5 | Depleted | 0.047976 | 18 | NA | 0 |

CRISPRi treatment: CRISPRi library treatment[71] condition in which enrichment test was performed and genes were grouped. "Enriched" genes had FDR ≤ 0.01 and $\log_2$ fold change ≥ 1, "Depleted" genes had FDR ≤ 0.01 and $\log_2$ fold change ≤ −1, "Responder" genes were either enriched or depleted with FDR ≤ 0.01. *CLR* clarithromycin, *INH* isoniazid, *STR* streptomycin, *EMB* ethambutol, *RIF* rifampicin, *VAN* vancomycin, *BDQ* bedaquiline, *LZD* linezolid.

BP vs. CRISPRi hypergeometric test *p* value: Significant one-sided hypergeometric test *p* value (*p* value ≤ 0.05) for overrepresentation of the 207 genes with significant Bayes Probability (BP) (FDR ≤ 0.05) among CRISPRi knockdown strains under various treatment conditions.

n Sig. BP genes: Number of genes with significant BP that are also either enriched or depleted CRISPRi knockdown strains.

OSR vs. CRISPRi hypergeometric test *p* value: Significant one-sided hypergeometric test *p* value (*p* value ≤ 0.05) for overrepresentation of the 159 genes within the Oxidative Stress Response (OSR) network among CRISPRi knockdown strains under various treatment conditions.

n OSR genes: Number of genes within the OSR network that are also either enriched or depleted CRISPRi knockdown strains.

*For the complete version of this table, including the overrepresented Sig. BP genes and OSR genes, see Supplementary Data 2.

Our findings show that OxyR-deficient mutants are oxidative stress pre-adapted populations that potentiate rapid emergence of INH[R] strains in clinical Mtb. The Bayesian analysis of clinical Mtb samples from Vietnam recapitulated an established association between mutations in the *oxyR′-ahpC* intergenic region and INH

resistance. This association is supported by the well-established role of OxyR′ (the Mtb ortholog of OhrR) as a negative regulator of *ahpC*, which encodes alkyl hydroperoxide reductase—a key enzyme in detoxifying reactive oxygen and nitrogen species[87–90], associated with increased resistance to oxidative stress[91]. Furthermore, both depletion

## Oxidative Stress Response Network

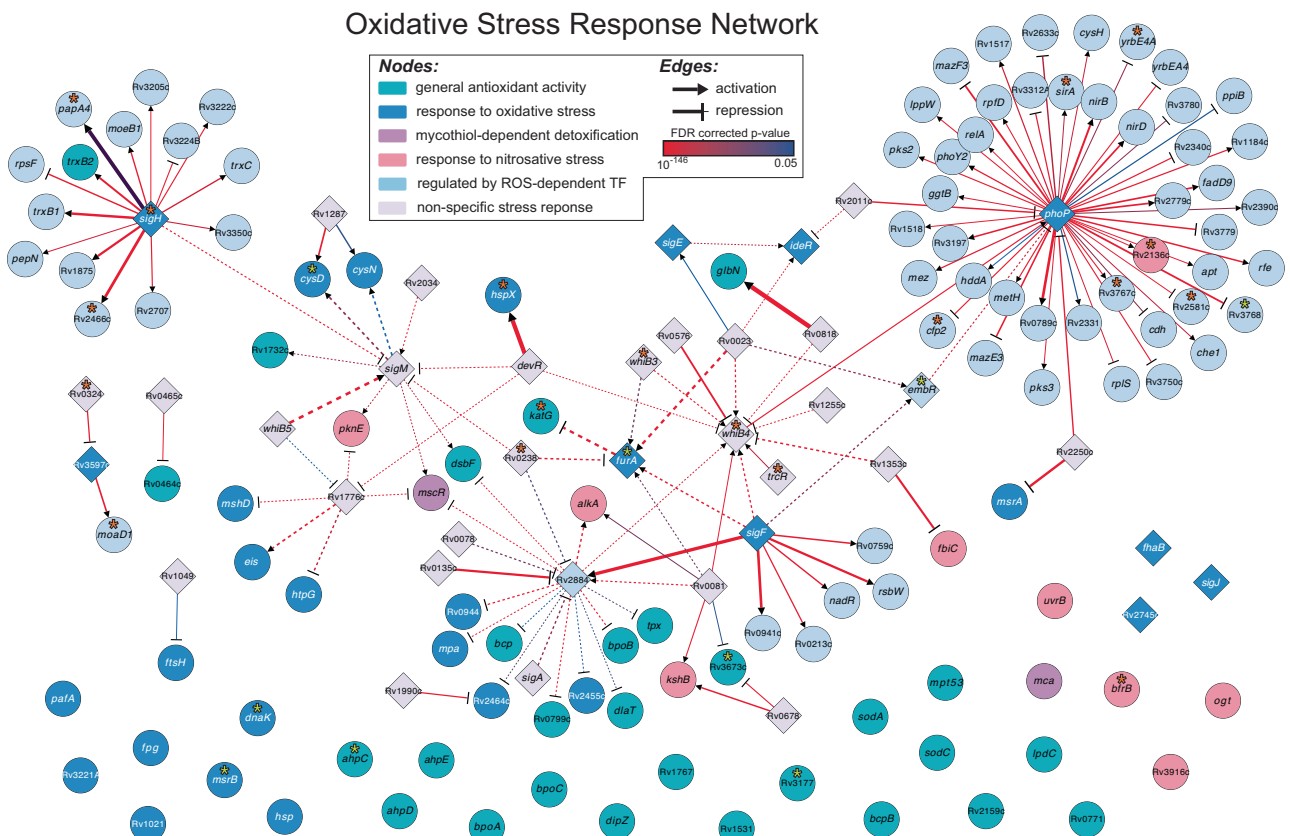

**Fig. 7 | Mutations in the oxidative stress response network in Mtb are significantly associated with the emergence of INH$^R$ strains.** Oxidative stress response network in Mtb inferred using TFOE and ChIP-seq datasets (see Methods for detailed description of how network was generated). The final network includes 159 nodes (genes) and 147 edges (TF regulatory influences). The node color represents the GO category. OSR genes without significant regulatory data were included as lone nodes at the bottom of the network. The shape of the edge end represents repression (bar) or activation (arrow). Bold edges represent interactions with evidence of TF influence (TFOE) and binding (ChIP-seq). The dashed edges represent interactions with only TFOE evidence. The color of the edge represents the TFOE FDR-corrected $p$ value, and the width of the edge represents the TFOE log$_2$ fold change in expression. Loci with significant BP for potentiating INH resistance are notated with an asterisk, with the color depending on if the significant probability corresponds to the gene itself (orange asterisk) or the upstream, intergenic region (yellow asterisk). Source data are provided as a Source Data file.

of the CRISPRi knockdown strain of *ahpC* in the genome-wide fitness screen in high-dose INH[71] as well as the complete absence of loss-of-function mutations in *ahpC* across clinically resistant Mtb strains[92] support the essentiality of *ahpC* for INH$^R$ TB infections. While previously it was believed that constitutive AhpC overexpression, through loss-of-function or regulatory mutations in *oxyR'*, was a compensatory mechanism[93] that appeared later to support INH resistance (primarily due to loss of function mutations in *katG*)[87], our findings suggest that *ahpC* overexpression mutations likely appeared first and potentiated the selection of high-level INH resistance mutations. In this regard, we note that exposure to host-derived ROS has likely fixed certain oxidative stress adaptations. For example, it is hypothesized that *oxyR* loss-of-function may have been an ancient adaptation of the *Mycobacterium tuberculosis* complex (MTBC) in the modern human host – a signature of the reductive evolution of Mtb's genome[94–96]. Rather than contradicting this, our findings suggest that such pre-adapted backgrounds remain permissive to further regulatory and metabolic variation that potentiates resistance acquisition. Mutations within conserved oxidative stress pathways may therefore represent ongoing evolutionary adaptation to elevated stress within host microenvironments. An example of this is the PhoPR two-component system. While Mtb's stress sensor domain of *phoR* is reported to be under ongoing positive selection in response to different host microenvironments, *phoP* is highly conserved due to its crucial regulatory role in virulence and the OSR network[97,98]. Interestingly, the CRISPRi *phoP* knockdown

strain is significantly depleted across drug treatments and has been proposed as a new drug target to treat TB[98]. Mtb resides primarily within host macrophages, where it is subjected to sustained oxidative and nitrosative stress as part of the innate immune response[75,99]. In this hostile environment, bacteria with enhanced oxidative stress defenses, such as those with derepressed *ahpC* expression, would be expected to have a selective advantage even in the absence of antibiotic pressure. Thus, *oxyR'-ahpC* promoter mutations are likely "pre-resistance" mutations[22] that incur minimal fitness cost under baseline conditions but prime Mtb populations for rapid acquisition of INH$^R$ with subsequent selective pressures during INH treatment.

The significant enrichment of oxidative stress-related loci among genes with high BP for association with INH resistance, further underscores the point that constitutive activation of oxidative stress defenses is not merely correlated with resistance but mechanistically linked to it. Specifically, this finding suggests that mutations in multiple genes associated with management of oxidative stress may have acted as pre-resistance mutations that led to the emergence and spread of INH$^R$ strains in Vietnam. Mathematical modeling of antibiotic tolerance and resistance emergence supports this interpretation[22]. Specifically, through model simulations Levin-Reisman et al. showed that tolerance mutations–those that transiently allow survival under antibiotic stress without conferring high-level resistance–can arise more frequently and in a broader range of genes than direct resistance mutations, effectively serving as evolutionary stepping stones toward

resistance. Consistent with this theory, our findings show that, with over 150 genes, Mtb's OSR network provides a much larger target for acquiring LLRT mutations without major fitness costs compared to the few genes involved in canonical high-level drug resistance. In fact, the high ROS within the host microenvironment likely favors selection of oxidative stress ameliorating LLRT mutations creating fertile evolutionary ground for the subsequent selection of stable, high-level resistance. Additional support for this theory comes from our tolerance assays, which showed that the slow rate of killing of all Msm LLRT mutants by 100× MIC INH was immediately followed by regrowth within 60 hours of treatment initiation, likely due to the rapid emergence of high-level INH$^R$ mutants. Together, these findings support a model in which oxidative stress first selects for LLRT mutations within the OSR network, thereby accelerating the evolutionary trajectory toward fixed, high-level drug resistance. The experimental demonstration that brief exposure to sub-inhibitory oxidative stress (via low-dose cumene hydroperoxide treatment) nearly tripled the rate of evolution to high-level INH resistance in Msm further provides compelling evidence that the oxidative environment within the host may also actively promote drug resistance evolution in Mtb.

This finding has important implications for understanding the epidemiology of drug-resistant TB. Patients with active TB experience chronic inflammation, and the granulomas that characterize TB pathology are sites of intense oxidative stress[100–102]. Our data suggest that this hostile microenvironment may inadvertently select for or enrich mutants with enhanced oxidative stress defenses, such as those with *oxyR'* mutations, thereby creating a reservoir of pre-resistant bacteria primed to rapidly acquire full resistance upon antibiotic exposure. Furthermore, recent studies have demonstrated that heterogeneity in drug penetration within granulomas creates microenvironments where bacteria experience sub-inhibitory antibiotic concentrations[18], which may also enrich for LLRT mutants as we have shown, further compounding the selective pressure for pre-resistance.

Our finding that *ohrR* loss-of-function LLRT mutants in Msm acquire high-level INH resistance at > 6-fold increased rate, primarily through subsequent mutations in mycothiol biosynthesis genes, provides a mechanistic explanation for this phenomenon for one such pre-resistance mutation in Msm. Loss of mycothiol, a key antioxidant in mycobacteria, is typically detrimental due to increased sensitivity to endogenously generated oxidative stress[26,56], and is particularly problematic for slow growing mycobacteria, such as Mtb, which accumulate more oxidative damage per cell division[103]. However, when alternate mechanisms of defense against oxidative stress are constitutively upregulated (as in *ohrR* mutants), the fitness cost of *mshA* mutations is buffered, allowing these otherwise deleterious mutations to be retained. Importantly, *mshA* mutations block the activation of INH by KatG, thereby conferring resistance through reduced drug activation rather than target modification[46,57]. This two-step evolutionary trajectory, wherein oxidative stress-mitigating mutations precede and enable the selection of resistance-conferring mutations, may explain the rapid emergence of high-level resistance observed clinically.

The findings presented here have several important clinical and therapeutic implications for TB treatment and drug development. First, they suggest that monitoring for mutations in OSR genes, particularly *oxyR'-ahpC*, may serve as an early warning for populations at elevated risk of acquiring drug resistance. Mutations in these loci may represent actionable biomarkers that warrant more aggressive treatment regimens or closer monitoring for resistance emergence. Additionally, lineage-specific genomic and regulatory variation may further modulate oxidative stress pre-adaptation and resistance trajectories. For example, Lineage 3 strains exhibit elevated *ahpC* expression due to intergenic variation, which may alter the selective landscape for acquiring additional resistance-potentiating mutations[104]. Consistent with prior work showing lineage-dependent resistance rates[105], these differences likely influence both the probability and pathways of resistance acquisition. Second, therapeutic strategies aimed at disrupting oxidative stress buffering systems may synergize with existing antibiotics. For instance, compounds that inhibit AhpC or other oxidative defense enzymes[106,107] could potentially reverse the permissive environment created by pre-resistance mutations, thereby reducing the likelihood of resistance evolution. Third, these findings underscore the importance of ensuring adequate drug penetration and compliance in TB treatment. The prolonged periods of sub-inhibitory drug exposure that can occur due to granuloma heterogeneity or treatment non-compliance create conditions that favor both the enrichment of pre-resistant mutants and the subsequent selection of high-level resistance[18,108–110]. Thus, our findings provide further support for the importance of strategies to improve drug delivery to granulomas or to reduce treatment duration without compromising efficacy to help mitigate the risk of emergence of drug resistance.

Several limitations should be noted. In this study, fitness was defined as growth rate and biomass accumulation in vitro, which are standard proxies for resource utilization and competitive capacity in bacterial systems[93,111]. We acknowledge that resistance-associated mutations may impose additional fitness costs not captured by these measurements, including effects on pathogenesis or metabolism in host-relevant conditions. Future work incorporating alternative fitness readouts will be necessary to assess these dimensions.

Although *ohrR* is not directly conserved in Mtb, the loss of *oxyR* in the MTBC represents an ancestral oxidative stress adaptation. Our findings highlight how disruption of oxidative stress regulation – regardless of the specific regulator – can create genetic backgrounds permissive to resistance evolution through the selection of mutations in conserved stress-response pathway genes, such as *ahpC*, *sigH*, *phoPR*, and mycothiol-dependent detoxification. Msm serves here as a closely-related model organism to identify general principles governing evolution of resistance to anti-mycobacterial drugs, which we then independently evaluate in clinical Mtb populations. Wet lab experiments in Mtb are a critical next step to help confirm the roles of oxidative stress and sublethal antibiotic exposure in the acquisition of high-level drug resistance. While we focused here on INH as a proof-of-concept redox-activated prodrug, additional antibiotics with related activation mechanisms, such as ethionamide and nitroimidazoles, represent important future directions. Our proof-of-concept study with INH demonstrated rapid identification of LLRT mutants in a one-step selection experiment, establishing a tractable framework that can now be extended to investigate evolutionary trajectories through which Mtb acquires high-level resistance to other antibiotics, including redox-dependent compounds.

The dataset of 1578 Mtb isolates, while valuable for linking genotype to experimentally determined phenotype, represents a single geographic region and does not fully capture global diversity. Additionally, the Bayesian framework employed here cannot definitively distinguish causal mutations from those linked by genetic hitchhiking, nor can it fully resolve the temporal sequence of mutational events. Longitudinal studies tracking the emergence of oxidative stress-related mutations in patients over the course of treatment would be valuable for validating these findings.

In this study, while host-derived ROS was not directly manipulated in vivo, our independent Bayesian analysis on clinical Mtb isolates revealed significant enrichment of resistance-potentiating mutations within the OSR network. Importantly, genes known to be involved in Mtb's response to the host phagocyte oxidative burst, such as *katG*, *ahpC*, *sigH*, *sodA*, etc., were all implicated in this analysis[85,112,113]. This result supports relevance to the host context, even in the absence of direct in vivo selection experiments. Our findings also suggest that high BP genes, especially those involved in the OSR network, may have also potentiated gain of resistance to other anti-TB drugs. Future work should explore this finding further, specifically, whether similar pre-

resistance mechanisms operate for other anti-TB drugs and whether oxidative stress-mitigating mutations also potentiate resistance to drugs beyond INH. Given that many drugs act through generation of oxidative stress[114], the principles identified here may have broad applicability with regard to understanding how including such drugs in combination regimen may potentiate the evolution of multidrug resistance. Conversely, the development of adjunct therapies targeting oxidative stress defenses, combined with traditional antibiotics, represents a promising avenue for clinical intervention.

In conclusion, this study demonstrates that oxidative stress management is a critical determinant of antibiotic resistance evolution in Mtb. Mutations that enhance oxidative stress resilience, likely pre-existing in clinical populations due to selection by the host immune environment, create a genomic background that accelerates the acquisition of high-level drug resistance without fitness trade-offs. These findings challenge the traditional view of resistance evolution as driven primarily by antibiotic exposure and highlight the importance of considering the host environment in shaping emergence of multi-drug resistance. Ultimately, targeting oxidative stress pathways in the pathogen may represent a novel strategy for preventing or delaying the emergence of drug-resistant TB.

## Methods

### Bacterial strains and growth conditions
All Msm strains were derived from the parental mc²155 strain, provided to us by the Bhatt lab at Stanford University. Additional strains W.J. Wildtype (mc²155 parental strain), *mshA KO* (in-frame deletion of *mshA* via phage transduction), and *mshA comp*. (*mshA KO* complemented with episomal plasmid expression of functional *mshA*) were kindly provided by the Jacobs lab at the Albert Einstein College of Medicine. Msm was grown at 37 °C in Middlebrook 7H9 broth or 7H10 plates supplemented with 0.2% glycerol, 0.05% Tween-80 (liquid media), and 10% OADC, with aeration. Media for all species of bacteria was free of antibiotics unless otherwise noted.

### Enrichment and selection of Msm LLRT mutants
Biological replicates of Msm were picked off of agar and inoculated into ~6 mL of media and grown until the cultures reached mid-log phase (Optical Density at 600 nm ($OD_{600}$) 0.6-1.0). Then, aliquots from each replicate were diluted and plated on antibiotic-free agar to be used as untreated controls. Next, the $OD_{600}$ of each replicate culture was normalized to 0.2 into 6 mL media + 2× $IC_{50}$ antibiotic (INH). Cultures were then incubated for 16 h to allow pre-resistant mutants to be enriched while also preventing the culture from undergoing an entire doubling, on average. Cultures were then plated on agar with 2× $IC_{50}$ antibiotic and then positioned on top of an image scanner placed inside an incubator. ScanLag was performed by capturing an image of sets of plates every hour for ~72 h[23]. Images were then compiled and colonies were detected and their growth metrics were inferred using the ScanLag software, available at https://github.com/baliga-lab/Scanlag.git. Of the colonies grown in the presence of INH, dimensionality reduction with PCA was performed on the scaled measurements of time of appearance, growth rate, and maximum colony size and unsupervised clustering using k-means with $k = 3$ was performed to group phenotypically distinct colonies together. K-means clustering of the PCA using $k = 3$ was determined based on the within sum of square distance from each cluster center. Representative colonies from each cluster, along with untreated control colonies, were then picked and assayed.

### Dose response assay for fitness and $IC_{50}$ determination
Dose response assays were performed in 384-well black-wall plates (Greiner Microplate, 384 Well, Cat # 781097) and growth within each well was recorded as the $OD_{600}$ with a plate reader

spectrophotometer. Wells with cultures were seeded at an initial $OD_{600} = 0.01$ and blank wells were filled with sterile media. A $\log_2$-fold dilution series of antibiotic concentrations ranging from 0× MIC to roughly 4-10× MIC was aliquoted out to their corresponding wells. Shortly after untreated cultures reached stationary phase, the experiment was stopped and the data was processed in R (version 4.1.1)[115]. Briefly, the mean of each blank $OD_{600}$ at each time point was subtracted from the $OD_{600}$ of the wells containing cultures. The area under the curve, lag phase, and additional growth metrics were inferred with GrowthCurver (version 0.3.1)[116]. Relative fitness was calculated as the ratio of the area under the growth curve (AUC) of the strain being analyzed during uninhibited growth (no antibiotic) relative to the average AUC of all untreated or wildtype control strains. To determine $IC_{50}$ values, a log-logistic model was fit to the dose response data using the drc package in R (version 3.0-1)[117].

### Tolerance assays
Twelve biological replicates of each tested strain were inoculated in fresh 7H9 media and grown overnight to mid-log phase ($OD_{600}$ of ~0.5) and diluted to $OD_{600}$ of 0.05, corresponding to ~$10^7$ CFU/mL in 3 mL of fresh 7H9. Isoniazid was added to each sample to a final concentration of 1 mg/mL (100× wildtype MIC). Cultures were incubated at 37 °C shaking at 225 RPM and subsampled at the following time points: 0, 6, 24, 30, 48, and 72 h. Aliquots were serially diluted 1:5 in fresh antibiotic-free 7H9 and a spotting assay was performed by plating out 4 µL of each dilution on 7H10 agar in an OmniTray (Thermo Sci Cat #140156). Agar plates were then incubated at 37 °C for 2-3 days until colonies appeared, at which point they were imaged and colonies were counted for CFU determination.

### Fluctuation assays
12 biological replicates of each strain used in these experiments were inoculated and grown to mid-log phase ($OD_{600}$ of ~0.5-0.8) in the absence of antibiotics. $OD_{600}$ of each culture was measured and normalized into 200 µL of media in a 96-well plate such that each well contained no more than ~200 cells. The plate was incubated with shaking until cultures reached late-log phase ($OD_{600}$ of ~1.0). Growth was stopped before cultures entered stationary phase to prevent the emergence of mutations that arise as a result of nutrient starvation. 40 µl of culture was serially diluted and 4 µl of each dilution was spotted on agar to determine the CFU/mL (population size). 100 µl of remaining culture in each replicate was plated on agar with 50× MIC INH. After colonies appeared, they were counted and mutation rates were determined using the web-based FALCOR tool (https://lianglab.brocku.ca/FALCOR/) using the frequency method and the Ma-Sandri-Sarkar Maximum Likelihood Estimator (MSS-MLE) method[44]. For fluctuation assays including pre-treatment with 135 µM cumene hydroperoxide (Thermo Fisher Cat #L06866), 1× $IC_{50}$ (as determined by dose response assay) was added to log-phase cells in 7H9 minimal media (7H9 + 0.2% glycerol, 0.05% Tween-80) to prevent degradation of the cumene hydroperoxide by catalase in the media for 24 h before cultures were plated on agar with and without 50× MIC INH for counting CFUs and mutant colonies exactly as described above. Mutation rates were adjusted for the difference in the number of generations the treated and untreated cells underwent before plating.

### NADH/NAD+ ratio determination
NADH and NAD+ concentrations were measured using the Enzychrom NAD/NADH assay kit (Bioassay Systems), following the manufacturer's instructions (and adding a bead-beating step of 15 min before heating at 60 °C for 5 min to lyse the bacterial cells). The equivalent of 1 mL of $OD_{600}$ of 1.0 bacterial cells (~$10^8$ CFUs) were used to measure NADH and NAD+ concentrations. To capture bacterial cells in the same growth phase in the treated and untreated conditions, these

experiments were performed separately on different days. In the absence of isoniazid, log-phase cultures were directly OD normalized and processed. For the treated condition, log-phase cultures were normalized to $OD_{600}$ of 0.95 and $1\times IC_{50}$ isoniazid (3.0 µg/mL for WT Parental, 270 µg/mL for WT s1 (*ndh::I65M*), 700 µg/mL for WT L2 (*ndh::L47F*), 9.4 µg/mL for ohrR anc (*ohrR::P4\*fs*), and 770 µg/mL for ohrR L10 (*ohrR::P4\*fs, mshA::R11\*fs*)) was added and cultures were incubated at 37 °C shaking at 225 RPM for two hours before NADH and NAD+ quantification. $1\times IC_{50}$ isoniazid was used to minimize the impact of fitness differences between the strains in the measured NADH/NAD+ ratio. Final results included at least two replicates per strain.

### ROS assay
At least three biological replicates of each indicated strain were grown to mid-log phase ($OD_{600}$ of 0.5-1.0) in 7H9 minimal media (7H9 + 0.2% glycerol, 0.05% Tween-80) and then normalized to $8 \times 10^6$ cells/mL. 200 µl replicates were added to black-walled 96-well plates (Thermo Sci Cat #237107). The fluorescent dye 2′,7′-dichlorodihydrofluorescein diacetate (H2DCFDA, Ex/Em: ~492–495/517–527 nm, Thermo Fisher Cat #D399) was added to OD normalized cultures of all indicated strains to a final concentration of 10 µg/mL, plates were sealed with foil sealing tape, and incubated at 37 °C for 1 h. RFUs were measured in a BioTek Synergy (H4 Hybrid Reader) fluorescence plate reader at the indicated excitation and emission wavelengths and fold change was calculated for each strain with respect to the wildtype parental strain in which each strain was derived.

### Whole genome sequencing
Mid- to late-log phase cultures were harvested by centrifugation and genomic DNA was extracted and purified with the Qiagen DNeasy UltraClean Microbial Kit (Cat #12224). Libraries for sequencing were prepared with the Nextera XT DNA library preparation kit (Illumina, San Diego, CA) and pooled for paired-end sequencing on a NextSeq instrument.

### Identification of mutations in evolved and wildtype Msm strains
Raw fastq files generated from Illumina sequencing were processed and mutations were called using a custom variant calling pipeline (https://github.com/baliga-lab/bwa_pipeline). Variant calling was performed with GATK (version 4.3.0.0)[118], Varscan (version 2.4.3)[119] and bcftools from Samtools package (version 1.6)[120]. Variants identified by each caller were collated and filtered for variant frequency ≥10%. Variants called by at least two algorithms were included for further analysis, including annotation using SnpEff (version 5.1 d)[121]. The reference genome used for Msm was mc²155: NC_008596.

### Identification of mutations in public Mtb WGS data
We harnessed whole genome sequencing data from the Sequence Read Archive (SRA) of 1578 clinical Mtb samples from Ho Chi Minh City, Vietnam[65]. The function *fasterq-dump* of the SRA toolkit (https://www.ncbi.nlm.nih.gov/books/NBK242621/) was used to download the raw *fastq* files and convert them to paired-end *fastq* files[122]. Reads were aligned to the reconstructed ancestral Mtb genome[66] and mutations were called using the same custom variant calling pipeline as above, adapted from[67], designed to capture fixed (≥ 90% alternative allele frequency) and unfixed (≥ 10% and ≤ 90% alternative allele frequency) intergenic and protein-coding mutations within each sample. As done in ref. 67, samples with an average sequencing depth over 20× and a mapping rate of over 90% were used for downstream analyses. Additionally, we excluded synonymous variants and variants located in regions that are difficult to characterize with short-read sequencing, including repetitive regions of the genome, such as PPE/PE-PGRS family genes, pro-phage genes, insertions or mobile genetic elements[123]. Computationally predicted lineage subtypes for each sample were identified using the TB-Profiler tool (version 6.3.0)[124].

### Bayesian analysis of Mtb metagenomes
After calling variants and collecting drug susceptibility profiles of each Mtb genome included in the dataset, an occurrence matrix was generated for all loci with at least one mutation in at least 1% of samples to associate mutation occurrence with drug susceptibility. To do this, we took the genotypic and phenotypic profiles of each sample, and used a Bayesian framework to calculate the conditional probability of an isolate being INH drug resistant given a mutation in any locus in the genome. We applied the classic Bayes Theorem:

$$P(A \mid B) = \frac{P(B \mid A) * P(A)}{P(B)}$$

where $P(A)$ is the probability of being resistant to a particular antibiotic (i.e., proportion of all samples resistant to INH), $P(B)$ is the proportion of samples with a mutation in the locus of interest, and $P(B \mid A)$ is the proportion of samples predicted to be drug resistant that also harbor a mutation in the locus of interest. Contextualized, Bayes Theorem can we written as:

$$P\left( INH^R \mid locus^{mut} \right) = \frac{P\left( locus^{mut} \mid INH^R \right) * P(INH^R)}{P(locus^{mut})}$$

This expression can then be expanded to show how each component is calculated, enabling us to simplify the equation:

$$P\left( INH^R \mid locus^{mut} \right) = \frac{\left( \frac{n\,with\,locus^{mut}\,and\,INH^R}{n\,with\,INH^R} \right) * \left( \frac{n\,with\,INH^R}{n\,total\,isolates} \right)}{\frac{n\,with\,locus^{mut}}{n\,total\,isolates}}$$

where $n$ is the number of samples with WGS that correspond to that given category (i.e. *'n total isolates'* is the total number of samples in the dataset, *'n with $locus^{mut}$'* is the total number of samples with at least one mutation in the specific locus). This expression can be simplified by canceling out like terms to arrive at the final equation used to calculate Bayes Probabilities:

$$P\left( INH^R \mid locus^{mut} \right) = \frac{n\,with\,locus^{mut}\,and\,INH^R}{n\,with\,locus^{mut}} \equiv \frac{P(locus^{mut} \cap INH^R)}{P(locus^{mut})}$$

We performed the conditional probability calculation for each genic and intergenic locus in the Mtb genome and then assigned $p$ values to each probability by comparing the value to the distribution of Bayesian probabilities for all loci across > 5000 iterations of randomizing the mutation profiles of all samples in the dataset. $p$ values were then FDR corrected for multiple testing and a significant Bayes probability was determined to be any loci with an FDR-corrected $p$ value ≤ 0.05.

### Development of oxidative stress response network in Mtb
A list of genes was compiled based on gene ontology for functions related to the following GO categories: antioxidant activity, GO:0016209; cellular oxidant detoxification, GO:0098869; removal of superoxide radicals, GO:0019430; mycothiol-dependent detoxification, GO:0010127; response to oxidative stress, GO:0006979; response to nitrosative stress, GO:0051409. We assembled a draft network by identifying experimentally predicted interactions and binding sites from transcription factor over-expression (TFOE)[72] and chromatin immunoprecipitation sequencing (ChIP-seq)[73] datasets, which were further summarized in ref. 74. The TFOE dataset was first filtered to only include influences with an FDR-corrected $p$ value ≤ 0.05. Then, each of the 206 TFs included in the TFOE dataset were assessed by hypergeometric test to identify regulons enriched for genes associated with ROS/RNS stress. Regulons enriched for

oxidative stress-related functions (hypergeometric test $p$ value $\leq 0.05$) were retained (5 TFs, 493 nodes, 531 edges). The network was then filtered to only include genes from the GO analysis (7 TFs, 28 nodes, 28 edges). The predicted transcriptional regulators (based on the analyzed experimental data) of the TFs with enriched regulons (25 TFs, 26 nodes, 40 edges) were then also added. Next, to include high-confidence regulatory relationships for genes that were filtered out during the enrichment analysis, TF-target interactions for TFs or targets in the GO gene list with significant TFOE influences and ChIP-seq binding evidence (18 TFs, 97 nodes, 89 edges) were also included in the network. Lastly, genes among the 68 identified from GO analysis that were left out of the network due to insufficient regulatory data (28 genes, 3 TFs) were manually added back to the network as lone nodes. The final network contained 37 TFs, 159 nodes and 147 edges and was visualized with Cytoscape v3.10.3[125] and analyzed with the NetworkAnalyzer[126] tool to identify node outdegrees.

### Statistics & Reproducibility

No statistical method was used to predetermine sample size. Sample sizes were selected based on prior experience with similar experimental systems and standard practices in mycobacterial microbiology to ensure sufficient reproducibility and statistical power. Eight independent biological replicates with at least three technical replicates each were used in the initial one-step selection experiment. Due to the very large number of isolates screened, dose response assays for fitness and IC$_{50}$ determination were performed with at least two technical replicates across two independent biological replicates (four replicate assessments in total). For tolerance assays, 12 biological replicates with at least two technical replicates each were used for CFU quantification. Fluctuation assays with and without the cumene hydroperoxide pretreatment were performed using 12 biological replicates inoculated at low cell density, with CFUs quantified across three technical replicates before and after selection with high-dose antibiotic. For NADH/NAD+ quantification, three biological replicates were assessed for each strain. For ROS measurements, at least three biological replicates per strain were assessed across two technical replicates. These sample sizes were sufficient to ensure reproducibility across independent experiments and to capture consistent phenotypic differences between strains and conditions. The final dose response assay in wildtype and ohrR::P4*fs in cumene hydroperoxide was performed with three technical replicates for each of three biological replicates.

No data were excluded from the analyses except where samples were lost during experimental processing. Specifically, in Fig. 4a and Supplementary Fig. 2, only two biological replicates were recovered for wildtype treated with 1× IC$_{50}$ isoniazid and strain WT s1 untreated due to sample loss. Data from WT s1 and WT L2 untreated were aggregated due to their highly similar genomic backgrounds. WT s1 was not measured under treated conditions. All other NADH/NAD+ quantification experiments were performed with three biological replicates.

All experiments were independently replicated as described in the Methods and figure legends. Biological replicates represent independently grown cultures, and technical replicates represent repeated measurements within each biological replicate. Key findings were reproduced across independent experiments. The experiments were not randomized. For the computational analysis, statistical significance of Bayesian probabilities was assessed by comparison to a null distribution generated from >5000 iterations of randomized mutation profiles across samples. Resulting $p$ values were corrected for multiple hypothesis testing using false discovery rate (FDR) adjustment. The investigators were not blinded to allocation during experiments and outcome assessment. Measurements including optical density, colony-forming units, and fluorescence-based assays were quantitative and instrument-based, minimizing the potential for observer bias.

### Reporting summary

Further information on research design is available in the Nature Portfolio Reporting Summary linked to this article.

## Data availability

All raw data underlying the figures are provided as a Source Data file with this paper. Raw whole-genome sequencing data for *Mycobacterium smegmatis* strains generated in this study have been deposited in the Sequence Read Archive (SRA) under BioProject accession PRJNA1363199. All strains generated in this study are available for reuse and can be obtained from the corresponding authors upon request. Whole-genome sequencing data for previously published *Mtb* isolates used in this study are available through the SRA under BioProject accessions PRJNA1028637 and PRJNA355614. Individual SRA accession numbers for these isolates are provided in the Source Data file. The *Mycobacterium smegmatis* reference genome (NC_008596.1) is available from NCBI. The inferred *Mtb* ancestral reference genome (Green et al., 2023)[66] was used for alignment and variant calling. Large intermediate datasets (e.g., Bayesian null distributions) are available via Zenodo[127] at [10.5281/zenodo.19207254]. Source data are provided with this paper.

## Code availability

All custom code used to generate the results and figures reported in this study is publicly available through a Zenodo archive of a GitHub release (v1.2)[128] [https://doi.org/10.5281/zenodo.19355216]. Any recent changes will be reflected in the newest release of the GitHub repository [https://github.com/evanpepper/Mycobacterium_PreR]. The repository contains R Markdown scripts and associated resources required to reproduce all computational analyses and figures presented in this manuscript. Software dependencies and package versions are specified within the repository.

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

## Acknowledgements

We thank members of the Baliga lab, including Jake Valenzuela, Min Pan, Amardeep Kaur, and Julie Do for their critical discussions, feedback, and general lab resource support. We also thank the Aitchison lab for additional discussions and feedback. We thank Sean M. Gibbons for all of his feedback and guidance on the interpretation of the results in this manuscript. We thank Catherine Vilchèze and William Jacobs Jr. at the Albert Einstein College of Medicine for providing their *M. smegmatis* mc[2]155 wildtype, ΔmshA knockout, and pMV361::*mshA* complemented strains used in this study. We thank the Bhatt lab from Stanford University for providing us with the parental *M. smegmatis* mc[2]155 strain used in this study and from which evolved strains in this study were derived from. We thank the Molecular and Cell Core Facility of the Institute for Systems Biology. We acknowledge the authors of[71] for their permission to use their data in our analysis. The article is licensed under a Creative Commons Attribution 4.0 International License (http://creativecommons.org/licenses/by/4.0/) and no changes were made to the data used. Lastly, we thank the Molecular Engineering and Sciences Institute at the University of Washington for their support. This work was supported by National Institutes of Health grants R01AI141953 (N.S.B) and R01AI128215 (N.S.B), with additional funding from the Bill and Melinda Gates Foundation INV-009322 (N.S.B) and INV-056403 (N.S.B). This work was also supported with funding from the Tuberculosis Research Unit (TBRU) grant U19AI162583, awarded to Jeffery S. Cox (UC Berkeley) and Thomas R. Hawn (University of Washington).

## Author contributions

E.P.-T. contributed conceptualization, data curation, formal analysis, investigation, methodology, software, validation, visualization, and writing – original draft preparation (primary). V.S. contributed conceptualization, investigation, and methodology. F.D.M. contributed conceptualization, methodology, resources, and software. S.L. contributed investigation. S.R. contributed investigation and software. W.H. contributed investigation. A.D.Z. contributed conceptualization, investigation, and methodology. W.W. contributed software. M.S. contributed data curation and resources. D.T.M.H contributed data curation. S.J.D. contributed resources and supervision. T.N.T.T. contributed funding acquisition and data curation. S.T. contributed investigation, resources, and software. J.D.A. contributed funding acquisition and supervision. M.L.A.-O. contributed conceptualization, methodology, resources, and supervision. N.S.B. contributed conceptualization, funding acquisition, methodology, project administration, resources, supervision, visualization, and writing – original draft preparation (support). Lastly, all authors contributed to writing – review and editing.

## Competing interests

The authors declare no competing interests.

## Additional information

¹Institute for Systems Biology, Seattle, WA, USA. ²Molecular Engineering and Sciences Institute, University of Washington, Seattle, WA, USA. ³Seattle Children's Research Institute, Seattle, WA, USA. ⁴Department of Pediatrics, University of Washington, Seattle, WA, USA. ⁵Department of Infectious Diseases, University of Melbourne at the Peter Doherty Institute for Infection and Immunity, Parkville, VIC, Australia. ⁶Pham Ngoc Thach Hospital for TB and Lung Disease, Ho Chi Minh City, Vietnam. ⁷Oxford University Clinical Research Unit, Hospital for Tropical Diseases, Ho Chi Minh City, Vietnam. ⁸Nuffield Department of Medicine, Centre for Tropical Medicine and Global Health, University of Oxford, Oxford, UK. ⁹Department of Biochemistry, University of Washington, Seattle, WA, USA. ¹⁰Department of Biology, University of Washington, Seattle, WA, USA. ¹¹Department of Microbiology, University of Washington, Seattle, WA, USA. ¹²Department of Civil & Environmental Engineering, University of Washington, Seattle, WA, USA. ¹³Lawrence Berkeley National Laboratory, Berkeley, CA, USA. ✉e-mail: mario.arrieta-ortiz@isbscience.org; nitin.baliga@isbscience.org

