## [Transparent Peer Review file · Nature Communications]

Host oxidative stress primes mycobacteria for rapid antibiotic resistance evolution

Corresponding Author: Dr Nitin Baliga

Version 0:

Reviewer comments:

Reviewer #1

(Remarks to the Author)

Host oxidative stress primes mycobacteria for rapid antibiotic resistance evolution

What are the noteworthy results?

It's known that ROS can enhance mutation rates and indeed has been shown through several studies that ROS can expedite the development of antibiotic resistance. Seminal study from James Collins lab kick started this work. Here the authors extend this by demonstrating that mutations that effect survival in ROS prime Mycobacteria to develop further mutations that cause high level resistance to antimicrobials. The relevance of this work to the actual pathogen Mycobacterium tuberculosis is supported by Bayesian analysis of M. tuberculosis clinical strains indicating that these mutations are associated with isoniazid resistance and also multi-drug resistance. Of course, as acknowledged by the authors this analysis cannot decipher the order that these mutations occurred. This will require sequential samples from the same patient. Isoniazid maybe particularly susceptible to this by virtue of the close relationship between redox and isoniazid mode of action. The authors reanalysis of a published CRISPRi screen suggests that this maybe applicable to multiple antibiotics with different modes of action but this would need confirmation in a future study.

This work has clinical significance as could identify patients harbouring strains which have a high chance of developing high level resistance. As isoniazid resistance is an important start in the journey to multi drug resistance in M. tuberculosis it doesn't actually matter if this phenomenon isn't relevant to other antibiotics.

I did find the paper quite a hard dense read though and could do with simplification to aid and attract readers and understanding of the study.

Will the work be of significance to the field and related fields? How does it compare to the established literature? If the work is not original, please provide relevant references.

This work although on mycobacteria is of relevance to other pathogens and will stimulate further studies.

Does the work support the conclusions and claims, or is additional evidence needed?

The only part which is more tentative is whether this is applicable to other antibiotics with different modes of action. The crispr suggests it could be but further studies in future papers will need to confirm this.

It would have been interesting to test antibiotics with similar modes of action/activation to isoniazid such as ethionamide and pretomanid.

Are there any flaws in the data analysis, interpretation and conclusions? Do these prohibit publication or require revision?

One of the threads through this paper that I struggle with is that the fitness costs that the author state is decreased growth rate. For Mtb is this a relevant fitness cost? Whilst it doesn't diminish the significance of the data I would disagree that the authors have challenged the dogma that antibiotic resistance doesn't come with a fitness cost when the only read out for this is growth rate. This is a pathogen that's virulence strategy is dominated by its ability to grow slowly or not at all. Whilst this does not diminish the significance of the paper this needs to be acknowledged and discussed in the article. The statement concerning "fitness costs" need to be softened. Also, when they refer to fitness they need to specifically state growth rate fitness as this is the only fitness they measure. Perhaps these snps have other "fitness costs" not related to growth rate in rich media in vitro.

L170. Further to this I don't agree that high level INH-R are unlikely to survive in the absence of antibiotic. Do you mean in vitro? Whats the evidence in vivo? Again this is only related to growth rate but if they have a reduced growth rate their survival maybe favoured within the host conditions. This needs to be qualified as it an assumption.

The authors should reference doi: 10.1128/JB.02252-14 which shows that deletion of

Is the methodology sound? Does the work meet the expected standards in your field?

Is there enough detail provided in the methods for the work to be reproduced?

Other comments

I also think the title is slightly misleading as they haven't demonstrated that directly that the host ROS is selecting for the mutations. This is inferred rather than directly shown.

Very Minor comments

L87. *M. smegmatis* needs to be in full the first time

Throughout the document: *M. smegmatis* and *M. tuberculosis* are sometimes abbreviated, sometimes not. This needs to be consistent

(Remarks on code availability)

I opened it but I'm not a coder! There were clear instructions though

Reviewer #2

(Remarks to the Author)

In this manuscript, Pepper-Tunick and colleagues investigate how mutations in oxidative stress response genes confer phenotypes tolerant to INH using *M. smegmatis* as a model for mycobacterial infections. These mutations appear to facilitate the subsequent acquisition of additional INH-resistant phenotypes through changes that impair mycothiol biosynthesis, mutations that are otherwise deleterious in different genetic backgrounds. In my view, the findings presented are highly relevant and compelling, as they address the emergence of non-canonical resistance in mycobacteria, a topic of considerable interest with significant knowledge gaps. Moreover, the data seem robust, and the experimental design appears sound and coherent. I have only a few comments:

I am unclear about the role of *fas1* mutations. They seem to be selected during the initial experiments with low-dose INH, yet subsequent testing indicates they do not enhance resistance or tolerance to INH. Why, then, were they selected in the first place?

The authors emphasize mutations in the *oxyR* - *ahpC* regulatory axis (*ohR* in *M. smegmatis*) and suggest that *oxyR*-deficient mutants may be predisposed to developing INH resistance. However, previous studies have shown that lineage 3 strains overexpress *oxyR* and *ahpC*, likely due to an intergenic mutation (<https://doi.org/10.1038/s41467-019-11948-6>). This suggests that the axis may be more active in L3 backgrounds. Therefore, could the mutations favored in *oxyR*-deficient mutants be less likely to occur in L3 strains?

The authors discuss how ROS within the host microenvironment may promote the selection of LLRT mutations, creating genetic backgrounds that facilitate high-level resistance. This pattern was observed in *M. smegmatis*, a free-living bacterium that, in principle, does not typically encounter ROS-rich host environments. In contrast, *M. tuberculosis* (and the entire MTBC) is an obligate pathogen that thrives under high ROS conditions, such as those found in granulomas. Thus, this selective pressure has likely shaped the evolutionary history of the group since its ancestor adopted a pathogenic lifestyle. The genetic background that promotes this low-level resistance in *M. tuberculosis* may be common MTBC genetic background/diversity, evolved for generations in high ROS concentration conditions. So, the mutations that the authors observed to raise in *M. smeg* may be fixed and common in *M. tb* genomes since then. Do these considerations align with the authors' interpretation?

P4-L108–L112: Did the authors include a growth control without INH supplementation to confirm that the fixation of these mutations was driven by low-dose INH selection pressure?

In this line, the authors state "...subjecting 8 replicate lines of log-phase *Msm* (mc2 110 155) to 2× IC50 (8.0 µg/mL) INH for 16 hours. Following the brief treatment, culture aliquots were plated on 7H10 agar with 2× IC50 INH and screened with ScanLag". But later on the methods "...the OD600 of each replicate culture was normalized to 0.2 into 6 mL media + 2× IC50 antibiotic (INH). Cultures were then incubated for 16 hours to allow pre-resistant mutants to be enriched while also preventing the culture from undergoing an entire doubling, on average. Cultures were then plated on agar with or without 2× IC50 antibiotic and then positioned on top of an image scanner placed inside an incubator." I am a bit confused as in the results the experiment seems to happen only in 2×IC50 conditions while in the methods the authors state that they also cultured in agar without antibiotics. Please, can you clarify this?

P5-L123–124: The sentence "...these mutants co-exist within a larger naïve wild-type mycobacterial population, even in the absence of antibiotic" would be better placed later in the manuscript, around page 9, where the authors specifically examine the coexistence of LLRT mutations with other variants in mycobacterial populations.

(Remarks on code availability)

The code used to generate the figures and analyze the data is available and can be used by the community. However the scripts rely in the installing of some libraries for proper running and this is not commented in the README file, while I suggest to include a 'dependencies' section to clarify this.

Reviewer #3

(Remarks to the Author)

The authors of this manuscript report about their findings, which link the brief exposure of mycobacteria to sublethal antibiotic concentrations or oxidative stress to the potential emergence of low level resistance against Isoniazid (INH), a standard first line anti-TB drug.

The authors have carried out their practical work by using *M. smegmatis* mc2 155, and by parallel bioinformatic analysis of 1578 *M. tuberculosis* strains from Vietnam.

The authors report that 6 of 40 cultures from colonies showing low level resistance and tolerance (LLRT) phenotypes had

mutations in genes linked to oxidative stress, including *ohrR* MSMEG_0448, *mfs1* MSMEG_2380, and *ntaA* MSMEG_6641. The authors then used a fluctuation tests and found that these LLRT mutants acquired high-level INH at an up to 6-fold higher rate, relative to the wildtype *M. smegmatis* strain. Based on these findings the authors formulated their hypotheses and linked their findings to mutations found in the genomes of the analyses 1578 *M. tuberculosis* strains.

Major comments:

The authors put much emphasis on the Loss-of-function mutations in *ohrR* (MSMEG_0448) corresponding to a gene encoding a transcriptional regulator of the MarR family, which in *M. smegmatis* shows the above described LLRT phenotype. To transpose these results onto *M. tuberculosis*, in table 1 it is mentioned that the orthologous gene of *ohrR* (MSMEG_0448) in *M. tuberculosis* is *oxyR'* (Rv2427A), a pseudogene. For this reviewer, it is thus unclear what impact the findings with the loss of function mutants in *ohrR* (MSMEG_0448) might have for *M. Tuberculosis*, which harbors already an inactive gene of *oxyR'*. In the discussion it is described that "OxyR-deficient mutants are oxidative stress pre-adapted populations that potentiate rapid emergence of INHR strains in clinical Mtb." But then all *M. Tuberculosis* strains must be like that, as *OxyR'* is a pseudogene in H37Rv with 100% identity in all *M. tuberculosis* strains tested by blast (mycobrowser.epfl.ch/; <https://blast.ncbi.nlm.nih.gov/Blast.cgi?>).

As such some wet lab experiments with *M. tuberculosis* would have been very helpful to confirm the conclusions drawn by the authors for *M. tuberculosis*.

(Remarks on code availability)

Version 1:

Reviewer comments:

Reviewer #1

(Remarks to the Author)

I am happy that you have addressed all my previous comments.

(Remarks on code availability)

I looked at this previously and they have updated in response to another reviewers comments

Reviewer #2

(Remarks to the Author)

Thanks for addressing all my comments.

(Remarks on code availability)

Reviewer #3

(Remarks to the Author)

The authors have discussed the different points raised by the reviewers and have revised their manuscript according to the reviewer's suggestion.

(Remarks on code availability)

POINT-BY-POINT RESPONSE TO REVIEWER CRITIQUES

Summary. We thank the editors and reviewers for their thoughtful and constructive feedback. In response, we have clarified key methodological details, refined our interpretation of oxidative stress pre-adaptation, and tempered language where conclusions are based on association rather than direct experimental demonstration. We expanded the Discussion to better frame lineage-specific oxidative stress regulation, the evolutionary context of *oxyR* loss-of-function, and the limitations of using *M. smegmatis* as a mechanistic model for *M. tuberculosis*. We also strengthened justification of our proof-of-concept focus on INH, incorporated additional references where appropriate, and improved code documentation to enhance reproducibility. Together, these revisions improve clarity and rigor while preserving the central conclusions of the study.

KEY: Green/Gray text: **Reviewer Comments**; Red/Blue text: **Authors' Response**

REVIEWER #1:

We sincerely thank you for your thoughtful and clinically grounded feedback, particularly your emphasis on careful interpretation of fitness costs and broader antibiotic applicability. Your comments helped us refine our framing and strengthen the clinical relevance of our findings. Please find below a point-by-point response to your comments.

Reviewer #1 (Remarks to the Author):

It's known that ROS can enhance mutation rates and indeed has been shown through several studies that ROS can expedite the development of antibiotic resistance. Seminal study from James Collins lab kick started this work. Here the authors extend this by demonstrating that mutations that effect survival in ROS prime Mycobacteria to develop further mutations that cause high level resistance to antimicrobials. The relevance of this work to the actual pathogen Mycobacterium tuberculosis is supported by Bayesian analysis of M. tuberculosis clinical strains indicating that these mutations are associated with isoniazid resistance and also multi-drug resistance. Of course, as acknowledged by the authors this analysis cannot decipher the order that these mutations occurred. This will require sequential samples from the same patient. Isoniazid maybe particularly susceptible to this by virtue of the close relationship between redox and isoniazid mode of action. The authors reanalysis of a published CRISPRi screen suggests that this maybe applicable to multiple antibiotics with different modes of action but this would need confirmation in a future study.

This work has clinical significance as could identify patients harbouring strains which have a high chance of developing high level resistance. As isoniazid resistance is an important start in the journey to multi drug resistance in M. tuberculosis it doesn't actually matter if this phenomenon isn't relevant to other antibiotics.

Critique 1.1: *I did find the paper quite a hard dense read through and could do with simplification to aid and attract readers and understanding of the study.*

Response: We appreciate the feedback from the reviewer regarding the density and complexity of the manuscript. However, considering neither of the other two reviewers mentioned this as a concern, we have decided to leave the bulk of the main text unchanged.

Critique 1.2: *The only part which is more tentative is whether this is applicable to other antibiotics with different modes of action. The CRISPRi suggests it could be but further studies in future papers will need to confirm this.*

Response: We agree that our strongest experimental support is for INH and that extension to other antibiotics should be framed cautiously. We also agree that the secondary analysis

(CRISPRi fitness profiling under multiple antibiotic conditions) should be compelling in that it suggests antibiotics of many different modes of action are relevant. As noted by the reviewer, this claim should be experimentally validated in future studies. We would like to direct the reviewer's attention to **Pg. 28, L557-L561** of the Discussion section, where we describe this limitation of our study.

Limitations and Future Directions: *"Our findings also suggest that high BP genes, especially those involved in the OSR network, may have also potentiated gain of resistance to other anti-TB drugs. Future work should explore this finding further, specifically, whether similar pre-resistance mechanisms operate for other anti-TB drugs and whether oxidative stress-mitigating mutations also potentiate resistance to drugs beyond INH."*

Critique 1.3: *It would have been interesting to test antibiotics with similar modes of action/activation to isoniazid such as ethionamide and pretomanid.*

Response: We agree that testing additional redox-activated prodrugs such as ethionamide and pretomanid would be highly informative. However, this work was meant to be a proof of concept using INH to demonstrate LLRT mutants can be rapidly identified in a one-step selection experiment, as opposed to requiring lengthy evolution experiments in the BSL-3 using Mtb. Now that we've established a robust workflow to enrich pre-resistance mutants, we have confidence in applying this approach directly to investigate how Mtb acquires high level resistance to a broader spectrum of antibiotics. Notably, prior studies demonstrate that mycothiol biosynthesis, stress-responsive SigH regulation, and redox homeostasis play direct roles in the activation and resistance mechanisms of both ethionamide and nitroimidazole drugs, reinforcing the biological relevance to our findings beyond INH (Cioetto-Mazzabò et al., 2023; Rawat et al., 2007; Vilchèze et al., 2008).

- **Edit (Discussion, Pg. 28, L535-541):** *"While we focused here on INH as a proof-of-concept redox-activated prodrug, additional antibiotics with related activation mechanisms, such as ethionamide and nitroimidazoles, represent important future directions. Our proof-of-concept study with INH demonstrated rapid identification of LLRT mutants in a one-step selection experiment, establishing a tractable framework that can now be extended to investigate evolutionary trajectories through which Mtb acquires high level resistance to other antibiotics, including redox-dependent compounds."*

Critique 1.4: *One of the threads through this paper that I struggle with is that the fitness costs that the author state is decreased growth rate. For Mtb is this a relevant fitness cost? Whilst it doesn't diminish the significance of the data I would disagree that the authors have challenged the dogma that antibiotic resistance doesn't come with a fitness cost when the only read out for this is growth rate. This is a pathogen that's virulence strategy is dominated by its ability to grow slowly or not at all. Whilst this does not diminish the significance of the paper this needs to be acknowledged and discussed in the article.*

The statement concerning "fitness costs" need to be softened.

Also, when they refer to fitness they need to specifically state growth rate fitness as this is the only fitness they measure. Perhaps these snps have other "fitness costs" not related to growth rate in rich media in vitro.

Response: We agree with the reviewer that fitness for any microbe, including Mtb, is multifaceted and cannot be fully captured by a single experimental readout. In this study, we use the term "fitness" in its conventional microbiological sense, referring specifically to growth rate and biomass accumulation *in vitro*, which remains a standard and widely applied proxy for fitness even in slow-growing pathogens (Andersson & Hughes, 2010; Melnyk et al., 2015). We also acknowledge that resistance-associated SNPs may impose additional fitness costs not measured here, including

effects on pathogenesis, metabolism, or host adaptation. These limitations are now explicitly described in the Discussion.

- **Edit (Discussion, Pg. 27, L519-525):** “*In this study, fitness was defined as growth rate and biomass accumulation in vitro, which are standard proxies for resource utilization and competitive capacity in bacterial systems (Andersson & Hughes, 2010; Melnyk et al., 2015). We acknowledge that resistance-associated mutations may impose additional fitness costs not captured by these measurements, including effects on pathogenesis or metabolism in host-relevant conditions. Future work incorporating alternative fitness readouts will be necessary to assess these dimensions.*”

Critique 1.5: L170. Further to this I don't agree that high level INH-R are unlikely to survive in the absence of antibiotic. Do you mean in vitro? Whats the evidence in vivo? Again this is only related to growth rate but if they have a reduced growth rate their survival maybe favoured within the host conditions. This needs to be qualified as it an assumption.

Response: We thank the reviewer for this important clarification and emphasize that our statement refers to *competitive fitness*, not absolute survival, in the absence of antibiotic pressure. While slow growth may favor treatment escape and survival within the host, multiple studies demonstrate that high-level INH resistance mutations – particularly *katG* loss-of-function mutations – incur substantial competitive disadvantages relative to the wild type drug-sensitive strain in mixed populations unless compensatory mutations arise (Andersson & Hughes, 2010; Dheda et al., 2017; Gagneux et al., 2006). Importantly, clinical and epidemiological data show that such resistance mutations remain rare in drug-naïve settings and predominantly emerge and persist following antibiotic exposure, consistent with purifying selection in the absence of treatment (Chiner-Oms et al., 2019; Ford et al., 2013). Recent work further supports this framework by highlighting that transmission success and population maintenance of INH resistant strains of Mtb are strongly constrained by fitness costs when antibiotic pressure is absent (Alame Emame et al., 2021). A few supporting references have now been added in the Introduction section on **Pg. 3, L67-68** and **L70**.

Critique 1.6: The authors should reference doi: 10.1128/JB.02252-14 which shows that deletion of

Response: We thank the reviewer for this suggestion and note that the referenced study (Saikolappan et al., 2014) is already cited at two points in the manuscript (**L134-L137, Figure 5 caption**), where its findings related to deletion of the *ohrR* gene are described in the context of antibiotic susceptibility and oxidative stress regulation, as shown below:

L134-L137: “*Loss of function mutations in ohrR, a transcriptional repressor of the ohr gene which encodes an organic hydroperoxide reductase, has been previously associated with low-level INH resistance in Msm (Meireles et al., 2022; Saikolappan et al., 2014).*”

Figure 5 caption: “**(C)** *Loss of function mutations in ohrR, such as ohrR::P4*fs, derepress ohr expression (Garnica et al., 2017; Saikolappan et al., 2014; Ta et al., 2011) resulting in significantly lower ROS levels and LLRT to INH (~3× IC₅₀).*”

Other comments

Critique 1.7: I also think the title is slightly misleading as they haven't demonstrated that directly that the host ROS is selecting for the mutations. This is inferred rather than directly shown.

Response: We respectfully disagree that the title is misleading. While host-derived ROS was not directly manipulated *in vivo*, our conclusions are supported not only by experimental data in Msm but also by an independent Bayesian analysis of 1,578 clinical Mtb isolates, which revealed significant enrichment of resistance-potentiating mutations within the oxidative stress response network. Importantly, genes known to be involved in Mtb's response to the host phagocyte

oxidative burst, such as *sodA*, *katG*, *ahpC*, *sigH*, and *whiB4*, were all implicated in this analysis (Chawla et al., 2012; Cioetto-Mazzabò et al., 2023; Master et al., 2002; Nambi et al., 2015). Together, these complementary datasets provide strong evidence that oxidative stress-associated selection pressures relevant to the host environment contribute to resistance evolution, justifying the framing used in the title.

- **Edit (Discussion, Pg. 28, L551-557):** *“In this study, while host-derived ROS was not directly manipulated in vivo, our independent Bayesian analysis on clinical Mtb isolates revealed significant enrichment of resistance-potentiating mutations within the OSR network. Importantly, genes known to be involved in Mtb’s response to the host phagocyte oxidative burst, such as katG, ahpC, sigH, sodA, etc., were all implicated in this analysis (Cioetto-Mazzabò et al., 2023; Master et al., 2002; Nambi et al., 2015). This result supports relevance to the host context, even in the absence of direct in vivo selection experiments.”*

Very Minor comments

Critique 1.8: *L87. M. smegmatis needs to be in full the first time. Throughout the document: M. smegmatis and M. tuberculosis are sometimes abbreviated, sometimes not. This needs to be consistent.*

Response: We thank the reviewer for noting this and have standardized the use of species names throughout the manuscript, defining *Mycobacterium smegmatis* and *Mycobacterium tuberculosis* in full at first mention and using consistent abbreviations thereafter.

Reviewer #1 (Remarks on code availability):

I opened it but Im not a coder! There were clear instructions though

Response: We thank the reviewer for checking the code availability and are glad the documentation and instructions were clear and accessible.

REVIEWER #2:

We greatly appreciate your positive and constructive evaluation of our work, as well as your insightful questions regarding lineage-specific oxidative stress regulation and methodological clarity. Your suggestions helped us improve our discussion of evolutionary context and improve transparency in both our experimental design and code documentation. Please find below a point-by-point response to your comments.

Reviewer #2 (Remarks to the Author):

In this manuscript, Pepper-Tunick and colleagues investigate how mutations in oxidative stress response genes confer phenotypes tolerant to INH using M. smegmatis as a model for mycobacterial infections. These mutations appear to facilitate the subsequent acquisition of additional INH-resistant phenotypes through changes that impair mycothiol biosynthesis, mutations that are otherwise deleterious in different genetic backgrounds. In my view, the findings presented are highly relevant and compelling, as they address the emergence of non-canonical resistance in mycobacteria, a topic of considerable interest with significant knowledge gaps. Moreover, the data seem robust, and the experimental design appears sound and coherent. I have only a few comments:

Critique 2.1: *I am unclear about the role of fas1 mutations. They seem to be selected during the initial experiments with low-dose INH, yet subsequent testing indicates they do not enhance resistance or tolerance to INH. Why, then, were they selected in the first place?*

Response: We thank the reviewer for this insightful question. *fas1* encodes a fatty acid synthase I (FAS-I) complex whose products (C₁₆ palmitate and C₂₆ fatty acids) feed directly into the FAS-II

pathway, for the manufacture of mycolic acids. The target of INH, *InhA* is a component of the FAS-II pathway, providing a plausible basis for the selection of *fas1* mutations under low-dose INH exposure (Bhatt et al., 2007). Although the *fas1* mutant did not ultimately increase the rate of acquisition of high-level resistance, it was likely selected or enriched due to subtle effects on lipid metabolism or cell wall composition under sublethal drug pressure that may have manifested in low level tolerance. Because this strain did not exhibit phenotypes relevant to resistance potentiation, we did not pursue it further and instead focused subsequent analyses on mutations that reproducibly enhanced resistance trajectories.

Critique 2.2: *The authors emphasize mutations in the oxyR - ahpC regulatory axis (ohR in M. smegmatis) and suggest that oxyR-deficient mutants may be predisposed to developing INH resistance. However, previous studies have shown that lineage 3 strains overexpress oxyR and ahpC, likely due to an intergenic mutation (<https://doi.org/10.1038/s41467-019-11948-6>). This suggests that the axis may be more active in L3 backgrounds. Therefore, could the mutations favored in oxyR-deficient mutants be less likely to occur in L3 strains?*

Response: We agree with the reviewer that lineage-specific regulation of the *oxyR'*-*ahpC* axis likely modulates selective pressures acting on resistance evolution. Indeed, Lineage 3 strains harbor intergenic variants that upregulate *ahpC* expression, consistent with enhanced oxidative stress defenses (Chiner-Oms et al., 2019), and this may alter the spectrum or probability of resistance-potentiating mutations relative to other lineages. As noted in our response to Critiques 2.3 and 3.1, we interpret *oxyR'* loss as one step in oxidative stress pre-adaptation, upon which additional regulatory or metabolic mutations may further modulate resistance trajectories. We agree that lineage-specific priming warrants explicit discussion that we now include in the updated Discussion section.

- **Edit (Discussion, Pg. 27, L500-505):** “Additionally, lineage-specific genomic and regulatory variation may further modulate oxidative stress pre-adaptation and resistance trajectories. For example, Lineage 3 strains exhibit elevated *ahpC* expression due to intergenic variation, which may alter the selective landscape for acquiring additional resistance-potentiating mutations (Chiner-Oms et al., 2019). Consistent with prior work showing lineage-dependent resistance rates (Ford et al., 2013), these differences likely influence both the probability and pathways of resistance acquisition.”

Critique 2.3: *The authors discuss how ROS within the host microenvironment may promote the selection of LLRT mutations, creating genetic backgrounds that facilitate high-level resistance. This pattern was observed in M. smegmatis, a free-living bacterium that, in principle, does not typically encounter ROS-rich host environments. In contrast, M. tuberculosis (and the entire MTBC) is an obligate pathogen that thrives under high ROS conditions, such as those found in granulomas. Thus, this selective pressure has likely shaped the evolutionary history of the group since its ancestor adopted a pathogenic lifestyle. The genetic background that promotes this low-level resistance in M. tuberculosis may be common MTBC genetic background/diversity, evolved for generations in high ROS concentration conditions. So, the mutations that the authors observed to raise in M. smeg may be fixed and common in M tb genomes since then. Do these considerations align with the authors' interpretation?*

Response: We agree with the reviewer that host-derived ROS has shaped the evolutionary history of *Mtb* and that some oxidative stress adaptations, such as loss of *oxyR*, are likely fixed in modern *Mtb* genomes (Deretic et al., 1997; Pagán-Ramos et al., 1998). Our interpretation is consistent with this view: rather than proposing *de novo* emergence of oxidative stress adaptations in *Mtb*, we suggest that host-adapted backgrounds provide a permissive baseline upon which additional regulatory or metabolic mutations further potentiate resistance acquisition. Notably, regulatory variation within oxidative stress pathways (e.g. *ahpC* promoter mutations, *sigH*, *phoPR*, mycothiol biosynthesis genes) remains under selection in clinical populations

(Buchmeier et al., 2003; Chiner-Oms et al., 2019; Rawat et al., 2002). The Msm experiments therefore serve as a mechanistic model illustrating how oxidative stress pre-adaptation lowers barriers to resistance, rather than as a direct evolutionary proxy. This framework is expanded upon in our response to Critique 3.1.

- **Edit (Discussion, Pg. 24-25, L418-432):** *“In this regard, we note that exposure to host-derived ROS has likely fixed certain oxidative stress adaptations. For example, it is hypothesized that oxyR loss-of-function may have been an ancient adaptation of the Mycobacterium tuberculosis complex (MTBC) in the modern human host – a signature of the reductive evolution of Mtb’s genome (Deretic et al., 1997; Pagán-Ramos et al., 1998; Veyrier et al., 2011). Rather than contradicting this, our findings suggest that such pre-adapted backgrounds remain permissive to further regulatory and metabolic variation that potentiates resistance acquisition. Mutations within conserved oxidative stress pathways may therefore represent ongoing evolutionary adaptation to elevated stress within host microenvironments. An example of this is the PhoPR two-component system. While Mtb’s stress sensor domain of phoR is reported to be under ongoing positive selection in response to different host microenvironments, phoP is highly conserved due to its crucial regulatory role in virulence and the OSR network (Chiner-Oms et al., 2022; Ryndak et al., 2008). Interestingly, the CRISPRi phoP knockdown strain is significantly depleted across drug treatments and has been proposed as a new drug target to treat TB (Ryndak et al., 2008).”*

Critique 2.4: P4-L108–L112: Did the authors include a growth control without INH supplementation to confirm that the fixation of these mutations was driven by low-dose INH selection pressure?

In this line, the authors state “...subjecting 8 replicate lines of log-phase Msm (mc2 110 155) to 2× IC50 (8.0 µg/mL) INH for 16 hours. Following the brief treatment, culture aliquots were plated on 7H10 agar with 2× IC50 INH and screened with ScanLag”. But later on the methods “...the OD600 of each replicate culture was normalized to 0.2 into 6 mL media + 2× IC50 antibiotic (INH). Cultures were then incubated for 16 hours to allow pre-resistant mutants to be enriched while also preventing the culture from undergoing an entire doubling, on average. Cultures were then plated on agar with or without 2× IC50 antibiotic and then positioned on top of an image scanner placed inside an incubator.” I am a bit confused as in the results the experiment seems to happen only in 2xIC50 conditions while in the methods the authors state that they also cultured in agar without antibiotics. Please, can you clarify this?

Response: We thank the reviewer for this important clarification. Antibiotic-free controls from each line were indeed included at the plating stage, but this was not described clearly in the Results section. Specifically, before the 16-hour liquid exposure to 2× IC₅₀ INH, culture aliquots from each line were plated on agar without antibiotic and tracked via ScanLag. Whole genome sequencing of colonies isolated from antibiotic-free plates (one colony per replicate line) revealed only the *ntaA_5* variant and did not detect the *ohrR* or *mfs1* mutations enriched on INH-containing plates. While the sample size from antibiotic-free plates was limited, these findings support our interpretation that LLRT mutations pre-exist in the naïve population and are enriched under sublethal INH exposure rather than being fixed independently of drug pressure. We have revised the manuscript to clearly distinguish the liquid pre-treatment and agar selection and to explicitly describe the antibiotic-free controls. In addition to these points, we also note that none of the wildtype-derived INHR mutants from the fluctuation test harbored an LLRT mutation, suggesting that the LLRT mutants we identified could only be enriched in sublethal antibiotic conditions.

- **Edit (Results, Pg. 4, L114-118):** *“In parallel, aliquots from each replicate line were plated on antibiotic-free agar prior to the 16-hour liquid INH exposure and screened with ScanLag separately as untreated controls. Altogether, the growth of 632 INH-treated colonies and*

537 untreated control colonies were detected and tracked with ScanLag. The 632 INH-treated colonies across the eight replicate lines...”

- **Edit (Results, Pg. 7, L151-155):** “Additionally, genome re-sequencing of one untreated control colony per line from antibiotic free-plates identified only the *ntaA_5* variant and did not detect *ohrR*, *mfs1*, or *fas1* mutations. Although the sampling depth of untreated controls was limited, this result supports the interpretation that LLRT mutations pre-exist in the naïve populations and are enriched under sublethal INH exposure.”
- **Edit (Methods, Pg. 29-30, L588-589; L597-598; L602):** “Then, aliquots from each replicate were diluted and plated on antibiotic-free agar to be used as untreated controls. Next, the OD_{600} of each replicate culture was normalized... Cultures were then plated on agar with $2\times IC_{50}$ antibiotic and then positioned... Of the colonies grown in the presence of INH, dimensionality reduction with PCA was performed... Representative colonies from each cluster, along with untreated control colonies, were then picked and assayed.”

Critique 2.5: P5-L123–124: The sentence “...these mutants co-exist within a larger naïve wild-type mycobacterial population, even in the absence of antibiotic” would be better placed later in the manuscript, around page 9, where the authors specifically examine the coexistence of LLRT mutations with other variants in mycobacterial populations.

Response: We thank the reviewer for this suggestion and agree that the later section provides a more detailed examination of LLRT coexistence. We intentionally introduce this statement earlier to contextualize the rapid detection of LLRT mutants in short-term selection experiments, which implies their presence within larger naïve populations. The later analysis then builds on this observation by providing direct evidence that LLRT mutations impose minimal fitness costs, offering an explanation for their coexistence and persistence. We believe that presenting this concept at both points – first as an observation and later with supporting evidence – improves the narrative and reinforces the consistency of the results.

Reviewer #2 (Remarks on code availability):

The code used to generate the figures and analyze the data is available and can be used by the community. However the scripts rely in the installing of some libraries for proper running and this is not commented in the README file, while I suggest to include a 'dependencies' section to clarify this.

Response: We thank the reviewer for this helpful suggestion. We agree that clearly listing required libraries and their version numbers is important for ensuring reproducibility and ease of use of our code. We have now added a dedicated “Dependencies” section to the GitHub README specifying all required packages and corresponding versions. See: https://github.com/evanpepper/Mycobacterium_PreR

REVIEWER #3:

We thank you for your careful reading and for highlighting important conceptual considerations regarding *oxyR* and the translatability of our experimental work to *Mtb*. Your comments prompted us to more precisely articulate the evolutionary framework underlying oxidative stress pre-adaptation and its implications for resistance emergence. Please find below a point-by-point response to your comments.

Reviewer #3 (Remarks to the Author):

The authors of this manuscript report about their findings, which link the brief exposure of mycobacteria to sublethal antibiotic concentrations or oxidative stress to the potential emergence of low level resistance against Isoniazid (INH), a standard first line anti-TB drug.

The authors have carried out their practical work by using *M. smegmatis* mc2 155, and by parallel bioinformatic analysis of 1578 *M. tuberculosis* strains from Vietnam.

The authors report that 6 of 40 cultures from colonies showing low level resistance and tolerance (LLRT) phenotypes had mutations in genes linked to oxidative stress, including *ohrR* MSMEG 0448, *mfs1* MSMEG 2380, and *ntaA* MSMEG 6641. The authors then used a fluctuation tests and found that these LLRT mutants acquired high-level INH resistance at an up to 6-fold higher rate, relative to the wildtype *M. smegmatis* strain.

Based on these findings the authors formulated their hypotheses and linked their findings to mutations found in the genomes of the analyses 1578 *M. tuberculosis* strains.

Major comments:

Critique 3.1: The authors put much emphasis on the Loss-of-function mutations in *ohrR* (MSMEG 0448) corresponding to a gene encoding a transcriptional regulator of the MarR family, which in *M. smegmatis* shows the above described LLRT phenotype. To transpose these results onto *M. tuberculosis*, in table 1 it is mentioned that the orthologous gene of *ohrR* (MSMEG 0448) in *M. tuberculosis* is *oxyR'* (Rv2427A), a pseudogene. For this reviewer, it is thus unclear what impact the findings with the loss of function mutants in *ohrR* (MSMEG 0448) might have for *M. Tuberculosis*, which harbors already an inactive gene of *oxyR'*. In the discussion it is described that "OxyR-deficient mutants are oxidative stress pre-adapted populations that potentiate rapid emergence of INHR strains in clinical Mtb." But then all *M. Tuberculosis* strains must be like that, as *OxyR'* is a pseudogene in H37Rv with 100% identity in all *M. tuberculosis* strains tested by blast (mycobrowser.epfl.ch/; <https://blast.ncbi.nlm.nih.gov/Blast.cgi?>).

Response: We agree that *ohrR* in Msm does not represent a direct orthologous mechanism in Mtb, where *oxyR'* is a pseudogene across the *Mycobacterium tuberculosis* complex (MTBC). Rather, we interpret the loss of *oxyR* as an ancestral host-adaptive event that pre-adapted pathogenic mycobacteria to oxidative stress, consistent with prior work (Buchmeier et al., 2003; Singh & Singh, 2009). Our findings extend this model by showing that oxidative stress pre-adaptation creates genetic backgrounds that are permissive to otherwise deleterious resistance mutations, a principle that does not depend on conservation of the regulator itself. Importantly, oxidative stress pathways – such as AhpC regulation, SigH-mediated stress responses, and mycothiol-dependent detoxification – are conserved and emerge as resistance-associated loci in our Bayesian analysis. Thus, *ohrR* serves as a tractable experimental proxy for studying how disruption of oxidative stress regulation accelerated resistance evolution, rather than implying a one-to-one genetic correspondence. This framework is expanded upon in our response to Critique 2.3 and in the updated Discussion section as shown below.

- **Edit (Discussion, Pg. 27-28, L527-531):** “Although *ohrR* is not directly conserved in Mtb, the loss of *oxyR* in the MTBC represents an ancestral oxidative stress adaptation. Our findings highlight how disruption of oxidative stress regulation – regardless of the specific regulator – can create genetic backgrounds permissive to resistance evolution through the selection of mutations in conserved stress-response pathway genes such as *ahpC*, *sigH*, *phoPR*, and mycothiol-dependent detoxification.”

Critique 3.2: As such some wet lab experiments with *M. tuberculosis* would have been very helpful to confirm the conclusions drawn by the authors for *M. tuberculosis*.

Response: We agree that wet lab experiments in Mtb are a critical next step, and emphasize that Msm serves here as a closely-related model organism to delineate general principles governing evolution of resistance to anti-mycobacterial drugs, which we then independently evaluate in

clinical Mtb populations. Now that we have established a framework for rapidly identifying LLRT mutants, we can confidently apply this approach to Mtb in the BSL-3, where determining the appropriate methodology would have been more challenging using a conventional laboratory evolution set up. We now acknowledge this limitation explicitly.

- **Edit (Discussion, Pg. 27, L531-535):** “Msm serves here as a closely-related model organism to identify general principles governing evolution of resistance to anti-mycobacterial drugs, which we then independently evaluate in clinical Mtb populations. Wet lab experiments in Mtb are a critical next step to help confirm the roles of oxidative stress and sublethal antibiotic exposure in the acquisition of high-level drug resistance.”

REFERENCES FOR POINT-BY-POINT RESPONSE TO REVIEWER CRITIQUES

- Alame Emame, A. K., Guo, X., Takiff, H. E., & Liu, S. (2021). Drug resistance, fitness and compensatory mutations in Mycobacterium tuberculosis. *Tuberculosis*, 129, 102091. <https://doi.org/10.1016/j.tube.2021.102091>
- Andersson, D. I., & Hughes, D. (2010). Antibiotic resistance and its cost: Is it possible to reverse resistance? *Nature Reviews Microbiology*, 8(4), 260–271. <https://doi.org/10.1038/nrmicro2319>
- Bhatt, A., Molle, V., Besra, G. S., Jacobs Jr, W. R., & Kremer, L. (2007). The Mycobacterium tuberculosis FAS-II condensing enzymes: Their role in mycolic acid biosynthesis, acid-fastness, pathogenesis and in future drug development. *Molecular Microbiology*, 64(6), 1442–1454. <https://doi.org/10.1111/j.1365-2958.2007.05761.x>
- Buchmeier, N. A., Newton, G. L., Koledin, T., & Fahey, R. C. (2003). Association of mycothiol with protection of Mycobacterium tuberculosis from toxic oxidants and antibiotics. *Molecular Microbiology*, 47(6), 1723–1732. <https://doi.org/10.1046/j.1365-2958.2003.03416.x>
- Chawla, M., Parikh, P., Saxena, A., Munshi, M., Mehta, M., Mai, D., Srivastava, A. K., Narasimhulu, K. V., Redding, K. E., Vashi, N., Kumar, D., Steyn, A. J. C., & Singh, A. (2012). Mycobacterium tuberculosis WhiB4 regulates oxidative stress response to modulate survival and dissemination in vivo. *Molecular Microbiology*, 85(6), 1148–1165. <https://doi.org/10.1111/j.1365-2958.2012.08165.x>
- Chiner-Oms, Á., Berney, M., Boinett, C., González-Candelas, F., Young, D. B., Gagneux, S., Jacobs, W. R., Parkhill, J., Cortes, T., & Comas, I. (2019). Genome-wide mutational biases fuel transcriptional diversity in the Mycobacterium tuberculosis complex. *Nature Communications*, 10(1), 3994. <https://doi.org/10.1038/s41467-019-11948-6>
- Chiner-Oms, Á., López, M. G., Moreno-Molina, M., Furió, V., & Comas, I. (2022). Gene evolutionary trajectories in Mycobacterium tuberculosis reveal temporal signs of selection. *Proceedings of the National Academy of Sciences*, 119(17), e2113600119. <https://doi.org/10.1073/pnas.2113600119>
- Cioetto-Mazzabò, L., Boldrin, F., Beauvineau, C., Speth, M., Marina, A., Namouchi, A., Segafreddo, G., Cimino, M., Favre-Rochex, S., Balasingham, S., Trastoy, B., Munier-Lehmann, H., Griffiths, G., Gicquel, B., Guerin, M. E., Manganelli, R., & Alonso-Rodríguez, N. (2023). SigH stress response mediates killing of Mycobacterium tuberculosis by activating nitronaphthofuran prodrugs via induction of Mrx2 expression. *Nucleic Acids Research*, 51(1), 144–165. <https://doi.org/10.1093/nar/gkac1173>
- Deretic, V., Song, J., & Pagán-Ramos, E. (1997). Loss of *oxyR* in Mycobacterium tuberculosis. *Trends in Microbiology*, 5(9), 367–372. [https://doi.org/10.1016/S0966-842X\(97\)01112-8](https://doi.org/10.1016/S0966-842X(97)01112-8)
- Dheda, K., Gumbo, T., Maartens, G., Dooley, K. E., McNerney, R., Murray, M., Furin, J., Nardell, E. A., London, L., Lessem, E., Theron, G., van Helden, P., Niemann, S., Merker, M., Dowdy, D., Van Rie, A., Siu, G. K. H., Pasipanodya, J. G., Rodrigues, C., ... Warren, R. M. (2017). The epidemiology, pathogenesis, transmission, diagnosis, and management

- of multidrug-resistant, extensively drug-resistant, and incurable tuberculosis. *The Lancet Respiratory Medicine*, 5(4), 291–360. [https://doi.org/10.1016/S2213-2600\(17\)30079-6](https://doi.org/10.1016/S2213-2600(17)30079-6)
- Ford, C. B., Shah, R. R., Maeda, M. K., Gagneux, S., Murray, M. B., Cohen, T., Johnston, J. C., Gardy, J., Lipsitch, M., & Fortune, S. M. (2013). Mycobacterium tuberculosis mutation rate estimates from different lineages predict substantial differences in the emergence of drug-resistant tuberculosis. *Nature Genetics*, 45(7), 784–790. <https://doi.org/10.1038/ng.2656>
- Gagneux, S., Long, C. D., Small, P. M., Van, T., Schoolnik, G. K., & Bohannon, B. J. M. (2006). The Competitive Cost of Antibiotic Resistance in Mycobacterium tuberculosis. *Science*, 312(5782), 1944–1946. <https://doi.org/10.1126/science.1124410>
- Master, S. S., Springer, B., Sander, P., Boettger, E. C., Deretic, V., & Timmins, G. S. (2002). Oxidative stress response genes in Mycobacterium tuberculosis: Role of ahpC in resistance to peroxynitrite and stage-specific survival in macrophages. *Microbiology*, 148(10), 3139–3144. <https://doi.org/10.1099/00221287-148-10-3139>
- Melnyk, A. H., Wong, A., & Kassen, R. (2015). The fitness costs of antibiotic resistance mutations. *Evolutionary Applications*, 8(3), 273–283. <https://doi.org/10.1111/eva.12196>
- Nambi, S., Long, J. E., Mishra, B. B., Baker, R., Murphy, K. C., Olive, A. J., Nguyen, H. P., Shaffer, S. A., & Sasseti, C. M. (2015). The Oxidative Stress Network of Mycobacterium tuberculosis Reveals Coordination between Radical Detoxification Systems. *Cell Host & Microbe*, 17(6), 829–837. <https://doi.org/10.1016/j.chom.2015.05.008>
- Pagán-Ramos, E., Song, J., McFalone, M., Mudd, M. H., & Deretic, V. (1998). Oxidative Stress Response and Characterization of theoxyR-ahpC and furA-katG Loci in Mycobacterium marinum. *Journal of Bacteriology*, 180(18), 4856–4864. <https://doi.org/10.1128/jb.180.18.4856-4864.1998>
- Rawat, M., Johnson, C., Cadiz, V., & Av-Gay, Y. (2007). Comparative analysis of mutants in the mycothiol biosynthesis pathway in Mycobacterium smegmatis. *Biochemical and Biophysical Research Communications*, 363(1), 71–76. <https://doi.org/10.1016/j.bbrc.2007.08.142>
- Rawat, M., Newton, G. L., Ko, M., Martinez, G. J., Fahey, R. C., & Av-Gay, Y. (2002). Mycothiol-Deficient Mycobacterium smegmatis Mutants Are Hypersensitive to Alkylating Agents, Free Radicals, and Antibiotics. *Antimicrobial Agents and Chemotherapy*, 46(11), 3348–3355. <https://doi.org/10.1128/aac.46.11.3348-3355.2002>
- Ryndak, M., Wang, S., & Smith, I. (2008). PhoP, a key player in Mycobacterium tuberculosis virulence. *Trends in Microbiology*, 16(11), 528–534. <https://doi.org/10.1016/j.tim.2008.08.006>
- Saikolappan, S., Das, K., & Dhandayuthapani, S. (2014). Inactivation of the Organic Hydroperoxide Stress Resistance Regulator OhrR Enhances Resistance to Oxidative Stress and Isoniazid in Mycobacterium smegmatis. *Journal of Bacteriology*, 197(1), 51–62. <https://doi.org/10.1128/jb.02252-14>
- Singh, A. K., & Singh, B. N. (2009). Differential Expression of sigH Paralogs during Growth and under Different Stress Conditions in Mycobacterium smegmatis. *Journal of Bacteriology*, 191(8), 2888–2893. <https://doi.org/10.1128/jb.01773-08>
- Veyrier, F. J., Dufort, A., & Behr, M. A. (2011). The rise and fall of the Mycobacterium tuberculosis genome. *Trends in Microbiology*, 19(4), 156–161. <https://doi.org/10.1016/j.tim.2010.12.008>
- Vilch ze, C., Av-Gay, Y., Attarian, R., Liu, Z., Hazb n, M. H., Colangeli, R., Chen, B., Liu, W., Alland, D., Sacchettini, J. C., & Jacobs Jr, W. R. (2008). Mycothiol biosynthesis is essential for ethionamide susceptibility in Mycobacterium tuberculosis. *Molecular Microbiology*, 69(5), 1316–1329. <https://doi.org/10.1111/j.1365-2958.2008.06365.x>